# BigMaQ: A Big Macaque Motion and Animation Dataset Bridging Image and 3D Pose Representations

**Lucas Martini**[1,2]    **Alexander Lappe**[1,2]    **Anna Bognár**[3]    **Rufin Vogels**[3]    **Martin A. Giese**[1]

[1]Hertie Institute, University of Tübingen    [2]IMPRS-IS    [3]KU Leuven

`lucas.martini@uni-tuebingen.de`

## Abstract

The recognition of dynamic and social behavior in animals is fundamental for advancing several areas of the life sciences, including ethology, ecology, medicine and neuroscience. Recent progress in deep learning has enabled an automated recognition of such behavior from video data. However, an accurate reconstruction of the three-dimensional (3D) pose and shape has not been integrated into this process. Especially for non-human primates, the animals phylogenetically closest to humans, mesh-based tracking efforts lag behind those for other species, leaving pose descriptions restricted to sparse keypoints that are unable to fully capture the richness of action dynamics. To address this gap, we introduce the **Big Ma**ca**Q**ue 3D Motion and Animation Dataset (`BigMaQ`), a large-scale dataset comprising more than 750 scenes of interacting rhesus macaques with detailed 3D pose descriptions of skeletal joint rotations. Recordings were obtained from 16 calibrated cameras and paired with action labels derived from a curated ethogram. Extending previous surface-based animal tracking methods, we construct subject-specific textured avatars by adapting a high-quality macaque template mesh to individual monkeys. This allows us to provide pose descriptions that are more accurate than previous state-of-the-art surface-based animal tracking methods. From the original dataset, we derive BigMaQ500, an action recognition benchmark that links surface-based pose vectors to single frames across multiple individual monkeys. By pairing features extracted from established image and video encoders with and without our pose descriptors, we demonstrate substantial improvements in mean average precision (mAP) when pose information is included. With these contributions, `BigMaQ` establishes the first dataset that both integrates dynamic 3D pose-shape representations into the learning task of animal action recognition and provides a rich resource to advance the study of visual appearance, posture, and social interaction in non-human primates. The code and data are publicly available at `https://martinivis.github.io/BigMaQ/`.

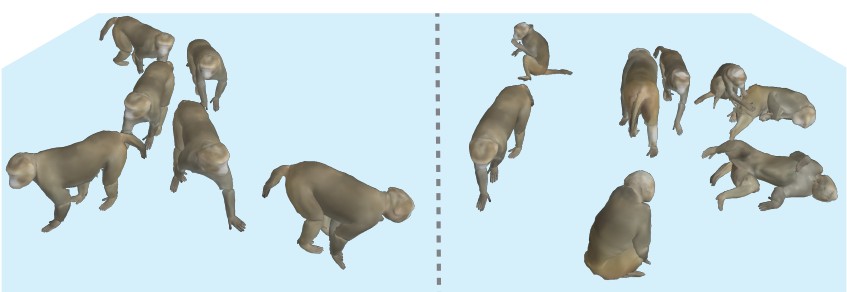

Figure 1: `BigMaQ`: A motion capture dataset with surface-based modeling for non-human primate behavior recognition. The dataset provides colored avatars that can be rendered from arbitrary views, including dynamic action reconstructions (left), rich individual and social interactions (right).

## 1 INTRODUCTION

Poses and shapes of human bodies have been effectively parameterized by 3D surface models using large amounts of data (Mahmood et al., 2019; Xu et al., 2020; Yu et al., 2020), enabling even the modeling of person-specific anatomy (Guo et al., 2022). The realization of a comparable representation for animals—despite its wide utility in the biological sciences—is hampered by the scarcity of accurate 3D data and motion recordings. As a result, surface-based models either come in a deformable generic shape space (Zuffi et al., 2017) or they are tailored specifically to individual species (Badger et al., 2020; An et al., 2023). Although surface descriptions exist for a variety of quadrupeds, behavioral research in species such as rodents, flies, or fish, typically relies on less detailed representations, most often in the form of 2D keypoints (Hsu & Yttri, 2021; Monsees et al., 2022; Lauer et al., 2022; An et al., 2023; Wu et al., 2024). The same holds for non-human primates (NHPs), particularly macaques, which provide critical insights that translate to humans (Gibbs et al., 2007; Cooper et al., 2022; Miller et al., 1996; Bethell et al., 2012), especially in visual and behavioral neuroscience (Rushworth et al., 2013; Hesse & Tsao, 2020).

In recent years, considerable progress has been made in capturing macaque body movements from images and videos (Bala et al., 2020; Labuguen et al., 2021; Yao et al., 2022). However, compared to humans and other animals, existing approaches lack the integration of detailed 3D shape representations of macaques. Since access to these animals is limited, most annotated videos are sampled either from wildlife scenarios (Ma et al., 2023; Brookes et al., 2024), where pose, if described at all, is represented by sparse 2D keypoints, or from captive environments, where movements are represented by 3D positions or keypoints (Bala et al., 2020; Marks et al., 2022; Martini et al., 2024; Matsumoto et al., 2025). Other approaches have incorporated synthetic data to couple 3D information with 2D image keypoints (Debnath et al., 2025). However, none of these efforts have included an accurate model of macaque body shape, nor have generic shape models adequately addressed this gap. Therefore, recent neurocomputational studies relied on human-based models to reconstruct 3D shape in monkey images (Yilmaz et al., 2025).

In this work, we address these limitations by augmenting the pose tracking of macaques with reconstructions of the complete 3D body surface, yielding more expressive models. In particular, we present `BigMaQ`, a large-scale macaque action dataset with realistic dynamic surface reconstructions covering more than 750 scenes of interacting individuals. A representative example is shown in Figure 1. By disentangling pose and shape, our dataset enables a detailed description of social interactions beyond previous keypoint-based approaches. In addition to the surface models, we provide extensive annotations including individual identities, segmentation masks, 2D keypoints, 16 calibrated viewpoints, per-frame poses, and action labels.

Our dynamic pose and shape representations describe actions across different body shapes, while also modeling finer movements such as hand rotations. By extending previous work in dynamic surface modeling of animals, we further incorporate a symmetric time loss, cropped differential rendering, and integrate texture into a sped-up optimization process, which makes the processing of the large amount of video data possible. The resulting dataset composed of dynamic and realistic avatars constitutes a unique resource, enabling detailed studies in behavior and perception for non-human primates. We are not aware of a similar dataset in other animal species combining realistic surface-based pose representations with action recognition in videos. In comparison with state-of-the-art mesh-based tracking approaches, our approach results in higher quality reconstructions both qualitatively and quantitatively. Finally, we create BigMaQ500, a benchmark of over 8k labeled videos with associated poses for action recognition in macaques. Using this dataset, we show that incorporating our pose descriptors along with video features from various foundation models not only substantially improves action recognition performance, but also outperforms commonly used descriptors in 2D and 3D.

## 2 RELATED WORK

While there exists a plethora of datasets for animal pose estimation, only a subset of these also provides annotations for behavior recognition. Fewer still predict poses with mesh-based models (see Table 1).

**Shape and Pose Reconstruction of Animals.** In animals, the simultaneous estimation of pose and shape from images can be split into model-free (Li et al., 2022; Yang et al., 2022a; Li et al., 2024) and model-based approaches (Zuffi et al., 2017; Wang et al., 2021b). While model-free approaches are promising for 3D shape retrieval from images or videos, model-based methods yield higher quality results for complex poses and real (non-synthetic) data (Rüegg et al., 2023). Therefore, we focus our discussion on these approaches. For many common animal species, model-based approaches rely on a generic quadruped shape space that was designed by scanning a variety of animal toy figurines, called SMAL (Zuffi et al., 2017). For dogs, this shape space has been fine-tuned to capture a greater diversity of species (Biggs et al., 2020; Rüegg et al., 2022). The shape model has also been extended to track other species deviating from its original shape space, like bears or camels (Biggs et al., 2019; Zuffi et al., 2018). However, SMAL's utility is limited for animals whose shape or naturalistic poses differ substantially from those of quadrupeds. To address this limitation, recent work has focused on customizing template meshes for specific animal groups. For example, Wang et al. (2021b) incorporated deformations from a template mesh of a bird, creating a unique shape space for different bird species. The model-based approach has also been extended to estimate shape and pose in videos using the SMAL shape model (Biggs et al., 2019; Sabathier et al., 2025), or by using template meshes of pigs, mice and dogs with multi-view recordings instead (An et al., 2023). Without access to accurate 3D data, even SMAL-based approaches rely on forming more informative priors by synthetic data or human-feedback (Zuffi et al., 2019; Xu et al., 2023). A recent line of work therefore extended the SMAL shape space to a wider range of mammal species using transformer models and synthetic data (Lyu et al., 2025b;a). Given that the pose space of NHPs differs substantially from other quadrupeds (Bala et al., 2020; Yao et al., 2022; Labuguen et al., 2021; Ma et al., 2023), these species ideally need rich motion and shape recordings similar to those in humans for realistic modeling of body and movement.

**Pose and Action Recognition in NHPs.** The aforementioned surface-based pose descriptions have only recently been introduced to describe animal poses in datasets (Xu et al., 2023; Lyu et al., 2025b). Traditionally, the pose is defined by 2D keypoints for both arbitrary species (Yang et al., 2022b; Ng et al., 2022) and NHPs in particular (Yao et al., 2022; Desai et al., 2023; Labuguen et al., 2021). Although datasets exist that enable the training of models for automated behavior recognition in conjunction with pose information (Marshall et al., 2021; Ng et al., 2022; Liu et al., 2023; Ma et al., 2023), the tasks of behavior recognition and pose estimation are typically treated as distinct training tasks (Luvizon et al., 2018; Duporge et al., 2024). Attempts to integrate pose information into action recognition (Luvizon et al., 2018; Choutas et al., 2018; Shah et al., 2022) have received less attention in recent research, largely due to rapid progress in model architectures for video classification, ranging from 3D convolutional (Carreira & Zisserman, 2017), two-stream (Feichtenhofer et al., 2019), to transformer-based networks (Arnab et al., 2021; Rajasegaran et al., 2023; Zhao et al., 2024). Further recent developments in this domain include large vision-language models (Zhong et al., 2025), or foundational encoders for video understanding (Zhao et al., 2024), which surpass the performance of species-specific models in behavior recognition of NHPs (Ma et al., 2023). Therefore, datasets that primarily focus on behavior recognition provide only minimal tracking annotations, such as bounding boxes (Brookes et al., 2024; Duporge et al., 2024; Fuchs et al., 2025; Vogg et al., 2025; Huang et al., 2025), or none at all (Chen et al., 2023).

Despite these advances in deep-learning based feature extraction from video, pose descriptions remain widely used to characterize, interpret and understand animal behavior (Mathis & Mathis, 2020; Bala et al., 2020; Hsu & Yttri, 2021; Lauer et al., 2022; Monsees et al., 2022; An et al., 2023; Matsumoto et al., 2025; Raman et al., 2025). In laboratory studies of key model species (e.g., rodents), higher-cost 3D keypoint representations are increasingly paired with behavioral annotations (Marshall et al., 2021; Patel et al., 2023). Beyond keypoints, 3D surface models have only recently been incorporated into the behavioral analysis of pigs (An et al., 2023), but as absolute, surface-level descriptors. A natural next step is, however, to include model-based pose representations into behavioral recognition, which has been shown to yield state-of-the-art performance in human action recognition (Rajasegaran et al., 2023). Apart from humans, we are not aware of any systematic dataset for learning such combined representations of poses and visual features in action recognition tasks.

## 3 BIGMAQ

The core feature distinguishing `BigMaQ` from other animal datasets, particularly those of NHPs, is that we provide accurate skeletal pose features of a 3D surface model in combination with action labels for large amounts of data. Unlike other datasets that provide shape and pose information for static images through manual annotation or synthetic data, we derive this information from video recordings of real behavior using multi-view markerless motion capture (see Table 1).

Table 1: Comparison of BigMaQ with existing pose estimation and action recognition datasets for NHPs, and animals in general. Species-specific datasets other than those containing non-human primates are not included. In the "Species" column, G denotes general, P primates, C chimpanzees, and M macaques. The column "Type" differentiates between 2D keypoint, 3D keypoint, and 3D-shape (3D-S) representations. The origin of this 3D data is further specified by S for synthetic and R for real recordings.

| Dataset | Species | Action Rec. | Video | Pose | | |
|---|---|---|---|---|---|---|
| | | | | Type | 3D-Data | Frame# |
| APT-36K (Yang et al., 2022b) | G | ✗ | ✓ | 2D | ✗ | 36,000 |
| AnimalKingdom (Ng et al., 2022) | G | ✓ | ✓ | 2D | ✗ | 33,099 |
| LoTE-animal (Liu et al., 2023) | G | ✓ | ✓ | 2D | ✗ | 35,000 |
| Animal3D (Xu et al., 2023) | G | ✗ | ✗ | 3D-S | S | 3,400 |
| AniMer (Lyu et al., 2025b) | G | ✗ | ✗ | 3D-S | S | 41,300 |
| OpenMonkeyChallenge (Yao et al., 2022) | P | ✗ | ✗ | 2D | ✗ | 111,529 |
| OpenApePose (Desai et al., 2023) | P | ✗ | ✗ | 2D | ✗ | 71,868 |
| ChimpACT (Ma et al., 2023) | C | ✓ | ✓ | 2D | ✗ | 16,028 |
| OpenMonkeyStudio (Bala et al., 2020) | M | ✗ | ✗ | 3D | R | 195,228 |
| MacaquePose (Labuguen et al., 2021) | M | ✗ | ✗ | 2D | ✗ | 13,083 |
| MacaqueMotionMonitor (Huang et al., 2025) | M | ✓ | ✓ | ✗ | ✗ | ✗ |
| **BigMaQ** (Ours) | M | ✓ | ✓ | 3D-S | R | 173,543 |

**Actions and Labels.** The video data for this study was captured using 16 high-precision color cameras at a frame rate of 40 frames per second (FPS), recording eight different male rhesus macaques (Macaca mulatta) in a neuroscientific laboratory (see Appendix A.1). The cameras were geometrically calibrated and synchronized, with light panels surrounding the enclosure, resulting in high-quality video footage of $2464 \times 2056$ pixels. More details are provided in Appendix A.2. From these recordings, 763 distinct actions were selected, featuring the monkeys engaging in either solitary behaviors or interactions with another conspecific. Except for feeding the monkeys with fruits or providing water, the recorded actions were not induced by humans but were based on the monkeys' own spontaneous behavior, and interactions with individuals in the same and neighboring enclosures.

We mapped individual behaviors to multiple action labels from a macaque ethogram (see Appendix A.3), since a single video can contain several simultaneous behavioral displays and is therefore inherently multi-label. To align these labels with existing action-tracking literature for NHPs (Ma et al., 2023), we grouped these actions into the following categories: (1) Locomotion; (2) Object

Interaction including also eating and drinking behavior; (3) Social Interactions between multiple individuals in a scene; and (4) Other kinds of behavior with more subtle social meaning. Individual action categories were aggregated from existing ethological studies in macaques, respectively (Altmann, 1962; Sade, 1973; Gunter et al., 2022). A more detailed description of the action categories is given in Appendix A.3, and their distribution across the dataset is shown in Fig 2.

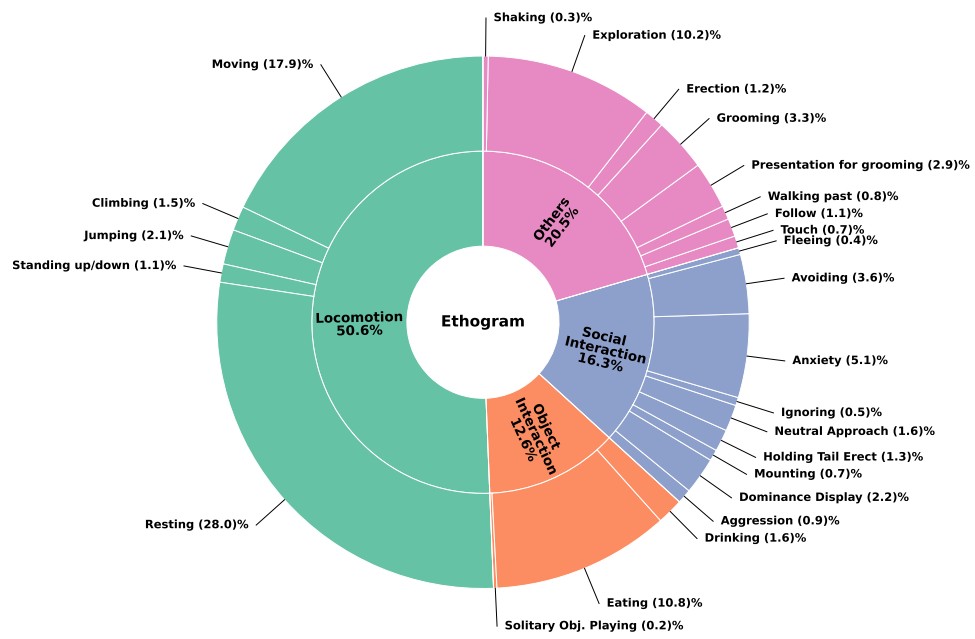

Figure 2: Distribution of the ethogram-aligned action category labels.

To align the surface model with the video data and individual specific action categories, we trained a detection model (YOLOv8, Jocher et al. (2023)) to predict monkey identities, and a widely used 2D keypoint estimator (An et al., 2023; Ma et al., 2023; Matsumoto et al., 2025), HRNet-W48 (Wang et al., 2021a), to predict 20 keypoints on cropped images. We included the end positions of hand and feet, thus extending usual marker set ups for NHPs (Labuguen et al., 2021; Yao et al., 2022) to hand and feet postures. An overview of the predicted keypoints is given in Appendix Figure 7. Annotations were generated for 306 monkey bodies or 3,712 images with internal software that projects already annotated 3D keypoints in remaining views to minimize manual labor. After training both models with this data (for model statistics see Appendix A.4), we generated keypoints (kps) and bounding boxes (bbox), each accompanied by confidence scores, for all 12,208 videos (764 × 16 views), and prompted a zero-shot foundational model (SAM 2, Ravi et al. (2024)) to retrieve segmentation masks. Additional details on annotation, quality checks and uncertainty propagation of the labels are described in Appendix A.5. In the following, we describe how the surface models are aligned with these label data and videos. An overview of the full pipeline and resulting datasets is shown in Figure 3.

**3D Mesh-Tracking.** For our macaques, we use a high-poly, artist-created template model (see Appendix Figure 9) composed of 10,632 vertices $\mathbf{V}_R \in \mathbb{R}^{N_V \times 3}$, and a low-poly version to fit the entire dataset, where $N_V = 3625$. These vertices are deformed and posed by an underlying rig via linear blend skinning (LBS); for more details, we refer the reader to Appendix B and to Loper et al. (2015). This rig consists of $N_J = 115$ joints in their template locations $\mathbf{J}_R \in \mathbb{R}^{N_J \times 3}$, where the influence of each joint on a single vertex is determined by the skinning weights $\mathbf{W} \in \mathbb{R}^{N_V \times N_J}$. To articulate the model, the pose parameters $\boldsymbol{\theta} \in \mathbb{R}^{3N_J}$ define the relative rotations for each joint in axis-angle representation, specifying how each joint is oriented with respect to its parent in the kinematic hierarchy. After LBS, the vertices are further rotated, scaled and shifted in global space by $\mathbf{R} \in \mathbb{R}^{3x3}$, $\gamma \in \mathbb{R}$, and $\mathbf{t} \in \mathbb{R}^3$, respectively. The entire process that computes posed vertices $\mathbf{V}_P$ can be written as

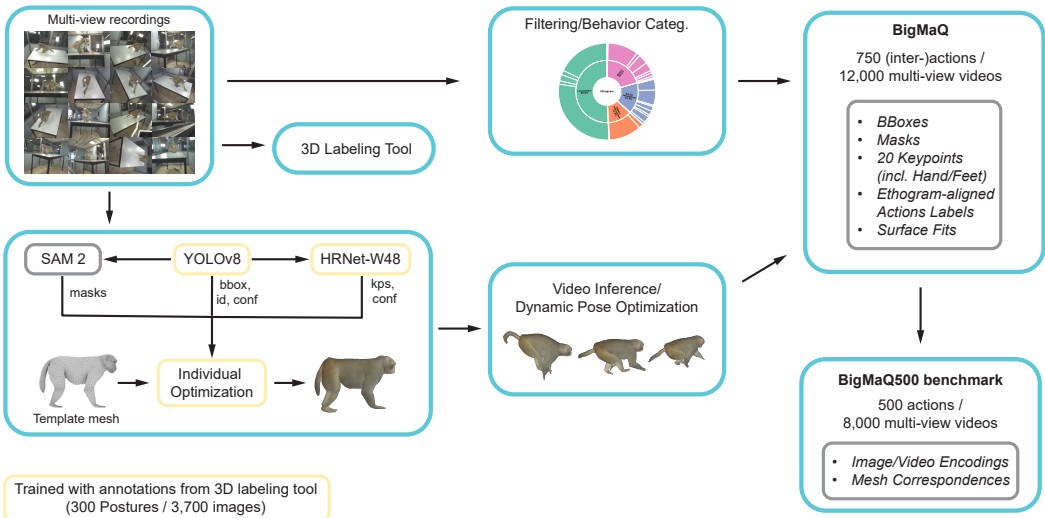

Figure 3: Pipeline overview for generating `BigMaQ` and the BigMaQ500 action–pose recognition benchmark. The 3D Labeling Tool provides annotations used to train the detection and keypoint models as well as to optimize subject-specific avatars. These optimized avatars are then combined with video-inferred labels to obtain dynamic pose reconstructions. BigMaQ500 includes all annotations available in `BigMaQ`, and additionally contains video encodings for more than 500 actions for which complete video-to-3D pose correspondences could be established.

$$\mathbf{V}_P = \gamma \cdot \mathbf{R} \cdot LBS(\boldsymbol{\theta}; \mathbf{V}, \mathbf{J}, \mathbf{W}) + \mathbf{t}. \tag{1}$$

The posed vertices and joints of the rig are then rendered as a mesh via differentiable rendering Ravi et al. (2020), resulting in predicted keypoints and silhouettes in the same perspective as our calibrated cameras. For the alignment of the model with the original videos, we define a frame-wise composite objective function

$$L(\Theta) = \lambda_P L_P + \lambda_b L_b + \lambda_{sm} L_{sm} + \sum_{\text{cam } c} \lambda_{kp} L_{kp}^c + \lambda_{sil} L_{sil}^c,$$

over the entire set of learnable parameters $\Theta = \{\boldsymbol{\theta}, \gamma, \mathbf{R}, \mathbf{t}, \boldsymbol{\alpha}, \boldsymbol{\xi}\}$. Instead of fixed joint positions, we use learnable bone length parameters $\boldsymbol{\alpha}$ as well as vertex offsets $\boldsymbol{\xi}$ to adapt the template mesh to individual specific differences. To this end, $L_P$ penalizes extreme joint rotations, $L_b$ ensures that adaptive bone lengths lie in a reasonable range, $L_{sm}$ handles smooth vertex deformations, and $L_{sil}$ and $L_{kp}$ align the articulated mesh with keypoints and silhouettes. The $\lambda_i$ are positive weighting coefficients. Since these adaptations and individual error terms have been discussed in prior work Badger et al. (2020); Rüegg et al. (2022), further details are given in Appendices B.1 and B.2. We focus in the following on the features that make our pipeline feasible for the entire dataset, and discuss contributions beyond most recent efforts (An et al., 2023).

**Large Data and Temporal Adjustments.** To reduce memory and computational load, we do not operate on full image frames to fit the surface model, but instead use cropped views for each frame and camera view, as given by our detection model (YOLOv8). We ensure a fixed maximum size for these cropped views and pass through the cameras sequentially in a batched format, which would be infeasible in high resolution. This allows us to define a temporal loss between all time points within a batch to enforce temporal consistency of the pose parameters. For rotations, we minimize angular velocities exploiting axis-angle rotations instead of Euclidean distances. Specifically, we define the angular velocity loss by the mean squared angular velocities across all joints and timesteps given as

$$L_{\text{ang}} = \frac{1}{(T-1)J} \sum_{n=1}^{T-1} \sum_{j=1}^{J} \left\| \boldsymbol{\omega}_j^{(n)} \right\|_2^2,$$

where $\boldsymbol{\omega}^{(n)}$ denotes the instantaneous angular velocity (see Appendix B.3 for more details). For global translation, however, we compute mean squared temporal differences in Euclidean space. Therefore, let $\mathbf{t}^{(n)} \in \mathbb{R}^3$ denote the global translation at the $n$-th timestep. Its smoothness loss is defined as the mean squared finite differences, yielding the combined temporal regularization objective

$$L_{\mathrm{T}} \;=\; L_{\mathrm{ang}}(\boldsymbol{\theta}_{:T}) + L_{\mathrm{ang}}(\mathbf{r}_{:T}) \;+\; \frac{1}{T-1}\sum_{n=1}^{T-1}\left\|\mathbf{t}^{(n+1)} - \mathbf{t}^{(n)}\right\|_2^2,$$

where $\mathbf{r}_{:T}$ denotes the axis-angle representation of the rotation $R$ for T timesteps. This loss is added to the per-frame loss defined in equation 1 for all timesteps within a batch. In practice, we use six cameras and batches of 80 timesteps with an overlap of 10 frames, using cropped frames with a maximum side length of 100 pixels. Generally, we first optimize the individual meshes on the labeled data used for training the detection and pose estimation models, yielding $\boldsymbol{\xi}$, $\boldsymbol{\alpha}$, and the colors per individual. Then, we optimize the poses of the individuals for each action following the optimization scheme given in Appendix B.4. That section also provides an overview of the resulting subject-specific meshes and the ranges of the learnable parameters estimated for each individual.

**Mesh Coloring.** To color the mesh, each vertex is associated with its own color vector, yielding the matrix $\mathbf{C} \in \mathbb{R}^{N_V \times 3}$ (RGB). Given posed vertices $\mathbf{V}_P$ and faces $\mathbf{F}$ connecting these vertices, a differentiable renderer $\mathcal{R}$ produces a color image $\hat{\mathbf{I}}$ under perspective camera $\Pi_c$ and lighting $\ell$:

$$\hat{\mathbf{I}}^{(c)} \;=\; \mathcal{R}(\Pi_c, \, \mathbf{V}_P, \, \mathbf{F}, \, \mathbf{C}, \, \ell)\,.$$

We estimate $\mathbf{C}$ per individual by minimizing a masked photometric objective on the training views:

$$\mathcal{L}_{\mathrm{phot}} \;=\; \sum_{\mathbf{p}\in\Omega}\mathbf{S}^{(c)}(\mathbf{p})\left\|\hat{\mathbf{I}}^{(c)}(\mathbf{p}) - \mathbf{I}^{(c)}(\mathbf{p})\right\|_2,$$

where $\mathbf{S}^{(c)}$ is the foreground silhouette mask, and $\Omega$ are image pixels. We ensure that $\mathbf{C}$ remains in the range $[0, 255]$ by applying a scaled sigmoid function to the learned parameters.

## 4 RESULTS

**Comparison to State of the Art.** We compare `BigMaQ` with the most recent approaches in both multi-view and single-view surface tracking. For multi-view evaluation, we consider MAMMAL (An et al., 2023) using our low-poly template model, which was also used to generate the pose data for all actions in `BigMaQ`. In addition, we evaluate AniMer+, a recent extension to SMAL that was designed for generic mammals (Lyu et al., 2025a;b). Qualitative results for four different actions, each performed by different individuals, are shown in Figure 4. AniMer+ fails to align the surface model with the image, estimating shapes that resemble entirely different animal species. For example, this is particularly apparent in the side-view shot where the mesh resembles more a lion or tiger. The template model in MAMMAL yields more plausible alignments, but it does not capture individual-specific differences and generally produces lower-quality fits compared to `BigMaQ`. Comparing the intersection-over-union (IoU) of mesh fits over time quantitatively confirms that our approach outperforms (An et al., 2023) in the multi-view setting (Table 2). To further evaluate the 3D alignment of our meshes relative to MAMMAL, we additionally report two standard metrics: the mean per-joint position error (MPJPE; Li et al. (2015); Zhou et al. (2016), in mm) and the mean per-joint temporal deviation (MPJTD or mean joint velocity error; Li et al. (2023); Pavllo et al. (2019), in mm/frame), which increases for jerkier or less temporally consistent keypoint trajectories. Similar to the IoU measured across viewpoint projections, our method achieves more accurate and smoother skeletal alignment for all actions shown. To demonstrate that these improvements are not limited to the examples in Figure 4, we also evaluated fits on single frames extracted from all single-subject actions in `BigMaQ` (Table 3). Across these frames, our method achieves higher IoU scores than previous approaches, and 3D skeletal alignment further confirms this advantage (26.907 mm for ours vs. 31.661 mm for MAMMAL). Evaluating the performance of `BigMaQ` over the entire time sequence of these single actions shows similarly strong performance (26.013 mm MPJPE and 8.463 mm/frame MPJTD). Further examples of dynamic surface captures are available in the Supplementary Video.

| **Image** | **BigMaQ-C** | **BigMaQ-M** | **MAMMAL** | **AniMer+** |

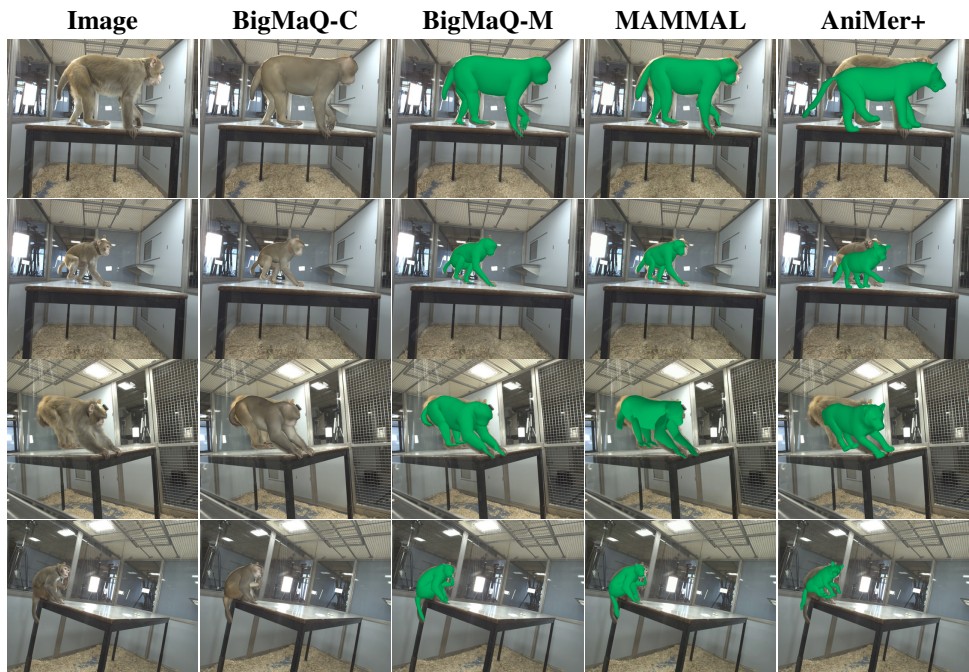

Figure 4: Qualitative comparison of different surface-tracking approaches for `BigMaQ`, illustrated for four actions of different individuals (rows). Only perspectives that were not used for multi-view optimization are shown. The second and third columns show the surface fits of `BigMaQ` with color texture (BigMaQ-C) and without (BigMaQ-M).

Table 2: Mean sequence-level surface reconstruction metrics for the actions shown in Figure 4. MPJPE is reported in mm and MPJTD in mm/frame. IoU scores are averaged over time and across cameras involved in the surface reconstruction.

| Action | IoU↑ | | MPJPE↓ [mm] | | MPJTD↓ [mm/frame] | |
|---|---|---|---|---|---|---|
| | MAMMAL | Ours | MAMMAL | Ours | MAMMAL | Ours |
| Walk | 0.771 | **0.883** | 23.493 | **20.402** | 9.961 | **6.875** |
| Food Picking | 0.756 | **0.855** | 25.521 | **16.489** | 13.362 | **9.062** |
| Branch Shake | 0.757 | **0.831** | 22.163 | **13.633** | 17.676 | **15.515** |
| Scratch | 0.722 | **0.843** | 28.843 | **20.481** | 8.498 | **4.422** |

While renderings of our low-poly meshes may occasionally reveal triangular artifacts on the back due to their lower vertex density, we additionally provide high-poly subject-specific meshes that can be rendered in the same poses without such artifacts. The tracked silhouettes from SAM 2, detections and keypoints are generally of high quality, but errors in these labels can impair the accuracy of our pose reconstructions, as illustrated in Appendix B.7. Such errors typically occur in more complex multi-individual scenes or when monkeys leave the volume covered by the cameras. Keypoint prediction errors, in particular, lead to stronger misalignments, since keypoints provide more informative constraints than silhouettes in the optimization process, as also demonstrated by a loss ablation for dynamic mesh fitting (Appendix B.6).

Table 3: Mean IoU across single-subject actions in `BigMaQ`, aggregating all viewpoints used in optimization for the first frame of the respective action.

| | BigMaQ | MAMMAL | AniMer+ |
|---|---|---|---|
| IoU | **0.844** | 0.714 | 0.591 |

Table 4: Comparison of action recognition based on pose representations $\theta$ and visual features including confidence intervals (mean $\pm$ std) for multiple training runs: Including pose features improves performance across visual backbones. The overall mean average precision (mAP) is further broken down by the categories defined in our ethogram: the subscripts L, OI, SI, and O denote Locomotion, Object Interaction, Social Interaction, and Others, respectively.

| Vision Model | Feat. | mAP | $\text{mAP}_\text{L}$ | $\text{mAP}_\text{OI}$ | $\text{mAP}_\text{SI}$ | $\text{mAP}_\text{O}$ |
|---|---|---|---|---|---|---|
| — | Pose | $43.5 \pm 1.4$ | $57.3 \pm 2.1$ | $\mathbf{54.4} \pm 1.3$ | $28.7 \pm 1.3$ | $47.4 \pm 4.0$ |
| ResNet50 | Vis | $34.3 \pm 0.5$ | $50.9 \pm 1.7$ | $38.6 \pm 1.3$ | $22.2 \pm 0.9$ | $35.9 \pm 0.9$ |
| | Vis+Pose | $\mathbf{44.0} \pm 0.8$ | $58.1 \pm 4.5$ | $53.7 \pm 1.0$ | $28.8 \pm 1.1$ | $\mathbf{48.5} \pm 2.3$ |
| ViT-base-cls | Vis | $32.9 \pm 0.7$ | $51.8 \pm 1.0$ | $37.4 \pm 1.5$ | $18.0 \pm 0.9$ | $35.9 \pm 1.1$ |
| | Vis+Pose | $\mathbf{44.0} \pm 0.1$ | $60.6 \pm 4.5$ | $50.2 \pm 3.0$ | $\mathbf{29.9} \pm 3.5$ | $47.1 \pm 3.9$ |
| DINOv2-base-cls | Vis | $37.8 \pm 0.7$ | $53.4 \pm 0.8$ | $49.9 \pm 0.8$ | $25.1 \pm 1.2$ | $37.8 \pm 1.4$ |
| | Vis+Pose | $41.8 \pm 2.6$ | $57.3 \pm 2.0$ | $52.9 \pm 1.4$ | $26.2 \pm 2.9$ | $45.5 \pm 4.9$ |
| ViT-base | Vis | $34.5 \pm 0.6$ | $52.3 \pm 0.9$ | $41.8 \pm 1.9$ | $20.0 \pm 1.3$ | $36.9 \pm 0.9$ |
| | Vis+Pose | $41.6 \pm 1.7$ | $61.5 \pm 1.8$ | $49.9 \pm 0.8$ | $28.4 \pm 0.9$ | $40.9 \pm 5.5$ |
| DINOv2-base | Vis | $40.4 \pm 1.7$ | $55.0 \pm 2.9$ | $51.0 \pm 0.6$ | $27.2 \pm 2.7$ | $42.1 \pm 1.2$ |
| | Vis+Pose | $41.4 \pm 1.7$ | $59.5 \pm 1.6$ | $51.4 \pm 0.6$ | $27.5 \pm 3.9$ | $42.0 \pm 0.8$ |
| TimeSformer | Vis | $31.9 \pm 1.2$ | $51.7 \pm 2.1$ | $34.8 \pm 2.7$ | $16.7 \pm 0.4$ | $35.7 \pm 1.8$ |
| | Vis+Pose | $42.6 \pm 1.3$ | $\mathbf{62.9} \pm 2.6$ | $49.6 \pm 1.8$ | $29.6 \pm 1.7$ | $41.9 \pm 3.9$ |
| VideoPrism-base | Vis | $38.3 \pm 0.2$ | $58.9 \pm 0.8$ | $49.1 \pm 0.2$ | $20.9 \pm 0.7$ | $40.7 \pm 0.9$ |
| | Vis+Pose | $43.8 \pm 2.9$ | $60.9 \pm 2.8$ | $51.1 \pm 1.2$ | $29.7 \pm 4.8$ | $46.3 \pm 5.6$ |

**Action Recognition.** To assess whether the pose vectors $\theta$ improve action recognition, we curated a subset of `BigMaQ` that retains only scenes in which for more than 95% of all timesteps and individuals successful pose reconstructions can be provided (see also Appendix A.5). We refer to this subset as BigMaQ500, which contains 511 actions or 8176 multi-view videos with long associated pose sequences. On this dataset, we trained three groups of transformer-based models, each combining video features with pose vectors. Videos and surface reconstructions were both sampled at 10 Hz, following previous work in humans (Rajasegaran et al., 2023). The first group of models used per-frame global embeddings: for ResNet50 (He et al., 2016), we extracted features just before the classification layer, whereas for vision transformer models—ViT-base (Dosovitskiy et al., 2021) pre-trained on ImageNet-21k (Ridnik et al., 2021) and the more recent DINOv2 (Oquab et al., 2023)—class tokens were used as global embeddings. The second group operated on patch tokens of the same vision transformer models, thereby retaining spatial information. The final group leveraged video encodings obtained from TimeSformer (Bertasius et al., 2021) and the foundation-scale VideoPrism model (Zhao et al., 2024). In addition, we trained a pose-only stream, absent of any visual input, to directly evaluate the contribution of pose features. Data splits to train these models included all individuals and action categories. More information on the model architecture and training is given in Appendix C. We evaluated all models using mean average precision (mAP), a standard metric in action recognition (Rajasegaran et al., 2023; Ma et al., 2023; Zhao et al., 2024). To account for the imbalance in categories and actions, we further report category-specific mAP, following (Ma et al., 2023). Our pose-only stream establishes a strong baseline, demonstrating that high-quality poses already provide sufficient cues to distinguish many actions (Table 4, second row). The combination of visual and pose features further improves performance, achieving a maximum overall mAP of 44.0 in two models. At the category level, different models yield the best results, but all models improve in action discrimination when augmented by our pose vectors. Similar to previous work in NHPs (Ma et al., 2023), social interactions remain the most challenging action category, underscoring the importance of pose features for modeling multi-individual behavior.

**The Effects of Pose Representation.** In the behavioral analysis of animals, the pose is usually not defined in terms of 3D generative parameters as in our vectors $\theta$, but rather utilizing 2D keypoints or 3D positional data (Mathis & Mathis, 2020; Bala et al., 2020; Lauer et al., 2022; Patel et al., 2023; An et al., 2023; Wu et al., 2024; Matsumoto et al., 2025). Therefore, we evaluate how the action recognition performance is influenced by different pose descriptions, including 2D and 3D keypoints, surface points, and our pose vectors represented as rotation matrices (see Table 5). Among these, the latter achieves the best results not only in pose-only stream models, but also when combined with visual embeddings based on class, spatial, and spatiotemporal tokens, indicating

that it is not the 3D information alone that drives recognition improvements. Instead, these results highlight the substantial benefits in behavior recognition of pose representations that construct 3D structure, rather than describing it. This perspective closely aligns with recent neuroscience studies that advocate incorporating generative approaches into recognition models (Yildirim et al., 2020; Yilmaz et al., 2025).

Table 5: Overall **mAP scores** for different pose representations in the `BigMaQ` action recognition task. Results are shown first for the pose-only stream, followed by ViT-base-cls, DINOv2-base, and VideoPrism visual features in combination with pose features. KP denotes keypoint positions in 2D and 3D, M the posed mesh's vertex positions, and Rot the matrix form of $\theta$ that consistently outperforms other pose descriptors.

| Features | — | ViT-base-cls | DINOv2-base | VideoPrism-base |
|---|---|---|---|---|
| 2D-KP | $35.6 \pm 0.3$ | $35.5 \pm 0.4$ | $40.2 \pm 1.2$ | $40.0 \pm 0.6$ |
| 3D-KP | $40.8 \pm 0.7$ | $34.4 \pm 1.1$ | $34.5 \pm 1.4$ | $34.0 \pm 2.0$ |
| 3D-M | $35.2 \pm 4.4$ | $34.4 \pm 2.4$ | $36.6 \pm 0.8$ | $36.0 \pm 3.3$ |
| 3D-Rot | $\mathbf{43.5} \pm 1.4$ | $\mathbf{44.0} \pm 0.1$ | $\mathbf{41.4} \pm 1.7$ | $\mathbf{43.8} \pm 2.9$ |

## 5 CONCLUSION

In this paper, we introduced `BigMaQ`, a video dataset combining markerless motion capture with 3D body surface and joint angle reconstruction, capturing a large variety of body movements and social interactions of macaques. Building upon previous methods of mesh-based animal tracking, we provide over 750 actions with rich annotations as multi-camera videos, individual detections, masks, keypoints, and most notably pose vectors per individual on a per-frame basis. Each individual monkey is associated with a subject-specific surface model, enabling a more accurate characterization of pose and body motion than basic image features or keypoints. These resources are unique in their composition and inspired by available motion capture data in humans. In making this dataset available, we hope to not only demonstrate the utility of more accurate descriptors for non-human primates, but also to promote a shift in action recognition to incorporate 3D knowledge about bodies into the action recognition models themselves. While we have demonstrated the capabilities of these descriptors for action recognition and tracking, the same representation can also be used for realistic, controlled animation. This paves the way for in-depth studies of perception and neural encoding of generative pose, shape, and social interactions in ecology and neuroscience.

**Limitations and Future Work.** Behavioral categories were labeled by only two researchers on this specific dataset, and extending category prediction to in-the-wild macaques or other NHPs would require broader consensus across behavioral experts. More challenging multi-individual scenes would benefit from view-consistent detection agreement, and SAM 2 silhouettes could be further improved by training a dedicated mask-quality discriminator. While our results demonstrate the advantages of pose-enabled action recognition primarily in multi-view scenarios, they are already informative for other species across the life sciences where 3D data or shape-spaces are available (Karashchuk et al., 2021; Marshall et al., 2021; An et al., 2023; Lyu et al., 2025b). Especially for quadrupeds, other approaches for in-the-wild image reconstructions do not rely on 3D data. While flexible, these models do not always yield faithful reconstructions (Li et al., 2024). In contrast, `BigMaQ` provides large-scale, high-quality 3D motion data for macaques. A natural next step is to derive a pose prior from this unique resource to regularize single-view reconstruction methods, as demonstrated for dogs in (Rüegg et al., 2022; Kearney et al., 2020). Such a prior could substantially improve generalization to complex poses and challenging in-the-wild imagery of NHP species.

**Reproducibility Statement.** We provide code and an anonymous URL to tracking labels in the Supplementary Materials. Additional dynamic surface estimates are included in the Supplementary Video. Furthermore, the Appendix provides a detailed account of the individual steps of the `BigMaQ` processing pipeline, including the recording setup, ethogram, keypoint annotation, quality control, pose/detector model statistics, loss terms, individual avatar statistics, ablations, optimization procedure, action-recognition training protocol, and computational requirements.

**Ethics Statement.** This work involves multi-view video recordings of captive macaques and the construction of a 3D motion dataset to support research on animal tracking and behavior understanding. All data collection was carried out in accordance with regional animal care guidelines and was approved by the relevant animal ethics committee. The macaques were part of an existing neuroscience facility, and no additional interventions, manipulations, or behavioral tasks were introduced specifically for this study.

The dataset contains only non-human primates and does not involve human subjects beyond the personnel recording the behavior. Care was taken to ensure that the video material does not reveal sensitive information about facility staff.

We acknowledge that automatic behavior recognition technologies may raise concerns related to surveillance or misuse if applied outside scientific and welfare contexts. We explicitly discourage any use of our methods or dataset for intrusive monitoring of humans or for applications that could negatively impact individual rights, privacy, or well-being. Similarly, we discourage deployments that could adversely affect animal welfare.

**Acknowledgements.** Funding was provided by the European Research Council (2019-SyG-RELEVANCE-856495). The authors thank the International Max Planck Research School for Intelligent Systems (IMPRS-IS) for supporting Lucas M. Martini. We are further grateful to S. Polikovsky for assistance with the motion capture hardware design and to the Max Planck Institute for Intelligent Systems and M. J. Black for providing the template macaque mesh model. We thank M. V. Carrera and M. U. Syed for support in video data annotation and processing, N. Taubert for mesh modeling, and I. Puttemans, A. Hermans, C. Ulens, S. Verstraeten, J. Helin, W. Depuydt, and M. De Paep for technical and administrative support.

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

## A  DATASET SPECIFICATIONS

### A.1  ANIMALS AND HUSBANDRY

The animals were housed in enclosures at the KU Leuven Medical School and experienced a natural day-night cycle. Each monkey shared its enclosure with at least one other cage companion and has access to toys and other enrichment. The monkeys received water and dry food ad libitum each day, supplemented by fresh fruits on friday and the weekend. Since the recorded monkeys are subjects in neuroscience studies, they were implanted with a plastic headpost and recording chamber. Both the animal care and experimental procedures adhere to regional (Flanders) and European guidelines and have been approved by the Animal Ethical Committee of KU Leuven.

### A.2  RECORDING SETUP

The 16 cameras used for the recording of the `BigMaQ` dataset (Victorem 51B163CCX) were precisely synchronized and recorded with video recorders (CORE2CXPLUS) from IO Industires, exploiting low-latency TTL signals. Together with the cameras also eight LED light panels (Strobo-Mini2, Norka Automation) were controlled by these TTL signals. This allowed to light the scene in synchrony with the individual camera shutters, and limiting the light intensity to avoid disturbing the animals. The cameras were mounted on 12 tripods with additional clamps as close as possible to the enclosures' glass in order to minimize reflections. An overview of the camera setup with corresponding camera IDs is shown in Figure 5. The enclosure measured $2.3 \times 3.1 \times 2.1$ m. All cameras were oriented toward the center of the enclosure, focusing on the table where the monkeys predominantly resided (0.9 m wide, 1.8 m long). Since the setup was not completely protected against mechanical perturbations, the cameras were re-calibrated at the start and end of each recording session by a ChArUco board manufactured from calib.io ($600 \times 400$ mm) with associated calibration software. To potentially reduce discrepancies between video recordings and colored meshes in post-production, we also captured color checkers (Calibrite ColorChecker Classic XL) and a white-balancing panel (Calibrite ColorChecker White Balance) at the beginning of each session. Videos were recorded as longitudinal data across eight daytime recording sessions. Actions were cut immediately on-site, in collaboration with the neuroscience experimenters, due to the vast amount of data generated in a single day.

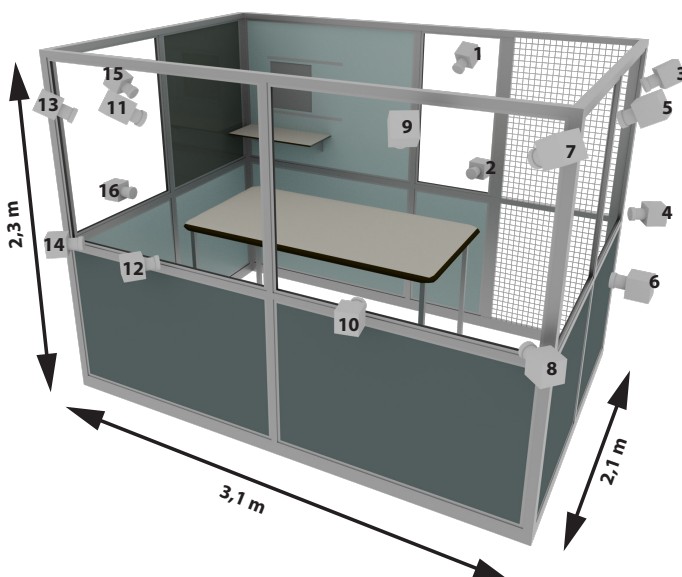

Figure 5: The recording setup of `BigMaQ` includes 16 cameras positioned around the monkey enclosure. The cameras were oriented toward the center of the space, specifically focusing the table area. Camera IDs are shown in black adjacent to each camera, with odd-numbered IDs indicating cameras providing a top-down view.

A.3 ETHOGRAM

We constructed an ethogram based on previous ethological studies in macaques (Altmann, 1962; Sade, 1973; Gunter et al., 2022), and grouped individual actions similar to recent action tracking in NHPs (Ma et al., 2023). The resulting actions and categories are as follows:

**Locomotion.** Patterns of self-initiated movement of an individual.

1. Moving — Horizontal movement, e.g., walking, running.
2. Climbing — Vertical movement, e.g., climbing up or down a structure.
3. Standing up/down — Raising or lowering the body from or into a bipedal stance.
4. Jumping — Moving the body by propelling it into the air and landing elsewhere.
5. Resting — Remaining stationary or only little movement, e.g., sitting or lying down, standing still on four legs.

**Object Interaction.** Direct physical interactions with inanimate stationary or movable objects using hands, feet, or mouth.

1. Solitary Object Playing — Non-social and non-goal-directed object manipulation (e.g., handling objects like poop or a cage).
2. Eating — Preparing and consuming food.
3. Drinking — Consuming liquids.

**Social Interaction.** Behavior that affects the behavior of at least another rhesus macaque. Loosely arranged from "Most aggressive", over "Neutral" to "Most submissive".

1. Aggression — Hostile interaction between two animals, with or without physical contact (includes push, chase, slap). In single monkeys can include branch shake or slapping the ground.
2. Dominance Display — Bouncing the body, bobbing the head, open-jawed/open-mouth gestures or conspicuous staring (often accompanied with beetled brows).
3. Mounting — Same-sex mounting interactions to assert dominance, reinforce status, or resolve conflicts.
4. Holding Tail Erect — "The tail is held up from the base, curled over backwards in the distal portion." (Altmann, 1962) Rank is communicated through the position of the tail.
5. Neutral Approach — Smacking lips, whipping tail, or walking toward another monkey in a neutral, non-hostile manner.
6. Ignoring — Continuation of a monkey's behavior even though it clearly perceived another monkey's behavioral pattern directed at it.
7. Self-directed behavior / Anxiety — Behaviors such as scratching, yawning, or self-grooming; can be associated with anxiety and can include exploratory actions.
8. Avoiding — Walking away, looking apprehensively, or grimacing toward another individual (e.g., opening the mouth widely, pulling back lips). Often accompanied by lateral flexion, stretched posture, or grimacing face to signal submission. Also includes avoiding to stare at a dominant individual close to it.
9. Fleeing — Running away from, or escaping another monkey.

**Others.** Other behaviors, if social, more subtle.

1. Touch — Approaching and touching another animal, distinct from grooming.
2. Follows — Persistent trailing of another animal, or continuous approach as the other moves away. Includes turning to gaze follow another monkey's actions.
3. Walking past — Passing by an individual; might be considered as first walking toward and then walking away from the other.

4. Presentation for grooming — Posture to solicit grooming from another animal, can include more than one animal or ambiguous intent.

5. Grooming — Picking through fur; one animal combing through another's hair, usually with hands, but sometimes with mouth.

6. Erection — Subtle social relevance, if any.

7. Shaking — Shaking the body or limbs.

8. Environment exploration — Moving and turning to inspect the environment.

Displayed behaviors and initial action labels were annotated on-site in consultation with neuro-physiologists to determine the monkeys' expressed behaviors. The refined ethogram labels were subsequently produced by the recording operator and reviewed during an initial labeling phase comprising more than 150 behavioral categories, together with an expert who works with these monkeys on a daily basis. The categorization of actions required approximately 18 hours.

While these labels serve as a description of the dataset and macaques display social behavior in laboratory environments (Franch et al., 2024), the recorded frequencies do not generally represent captive or wild-life monkey behavior. First, the videos were cut to show a variety of different and actions, and secondly, the monkeys' behavior could have been affected by the filming conditions. Moreover, the recorded male individuals were between 5 and 7 years old and can therefore capture the behavior of juveniles or females only to a limited extent.

### A.4 Detection and pose estimation

The image data to train both models comprises 3,712 images, where these frames include multiple viewpoints from poses of up to three interacting monkeys. Keyframes were selected based on distinct initial action categories and to maximize pose variability, while preserving a balanced distribution of individuals. The detection confusion matrix and precision-recall curve are displayed in Figure 6, showing good recognition performance across individuals.

We further report COCO results on the validation set based on default settings for training in the well-established toolbox from (Contributors, 2020; Yu et al., 2021) for pose estimation, and (Jocher et al., 2023) for detection.

Table 6: COCO evaluation metrics for HRNet-w48 at trained epoch 180. The model was trained with an 80%/10%/10% train/validation/test split. AP and AR denote average precision and average recall, respectively. The notation ' x' indicates evaluation at an IoU threshold of x (e.g., AP@0.5 = average precision at IoU = 0.5), while 'all IoU' refers to averaging across thresholds from 0.5 to 0.95 in steps of 0.05.

| Metric | Score |
|---|---|
| AP (all IoU) | 0.9268 |
| AP @ 0.5 | 0.9900 |
| AP @ 0.75 | 0.9664 |
| AR (all IoU) | 0.9355 |
| AR @ 0.5 | 0.9918 |
| AR @ 0.75 | 0.9696 |

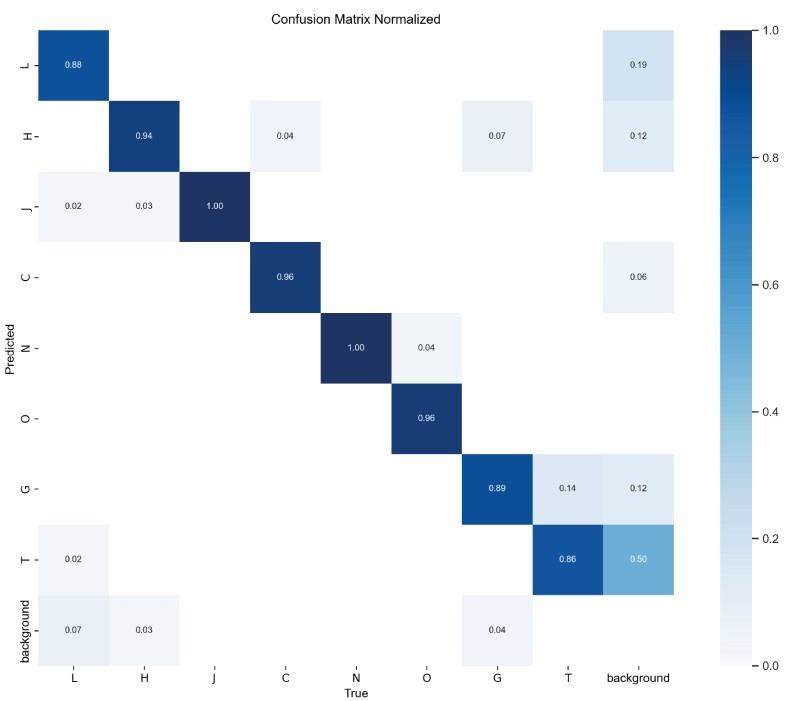

(a) Detection confusion matrix.

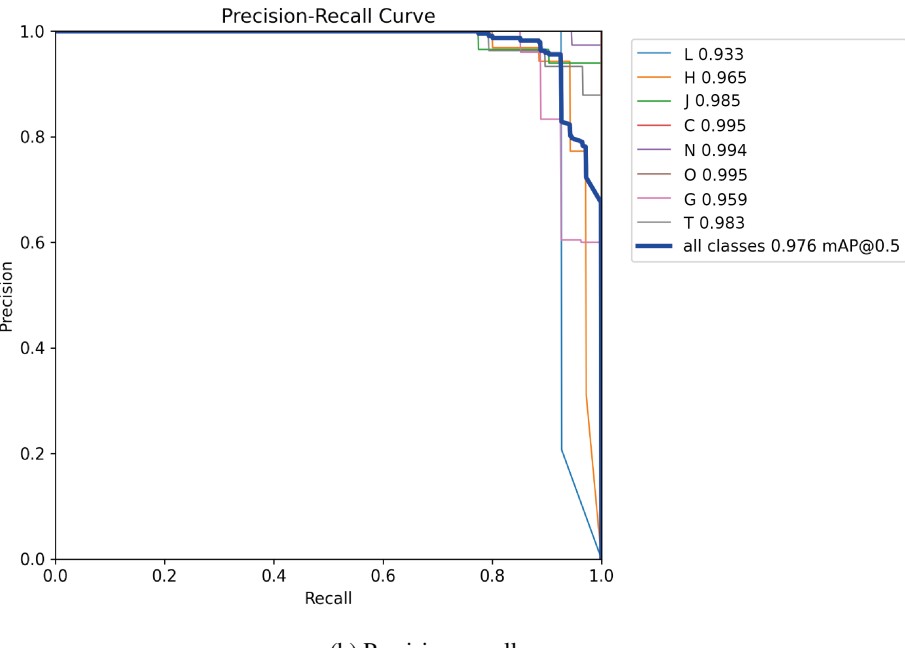

(b) Precision–recall curve.

Figure 6: Detection performance illustrated through the confusion matrix (top) and the precision–recall curve (bottom), demonstrating robust recognition across individuals.

Table 7: YOLOv8l model validation results at epoch 49. The model was trained with a 5% validation split for early stopping. mAP @ 0.5 indicates mean average precision at IoU = 0.5, while mAP @ 0.5:0.95 is the mean average precision averaged over IoU thresholds from 0.5 to 0.95 in steps of 0.05.

| Metric | Value |
|---|---|
| Precision (P) | 0.9526 |
| Recall (R) | 0.9561 |
| mAP @ 0.5 | 0.9762 |
| mAP @ 0.5:0.95 | 0.8538 |

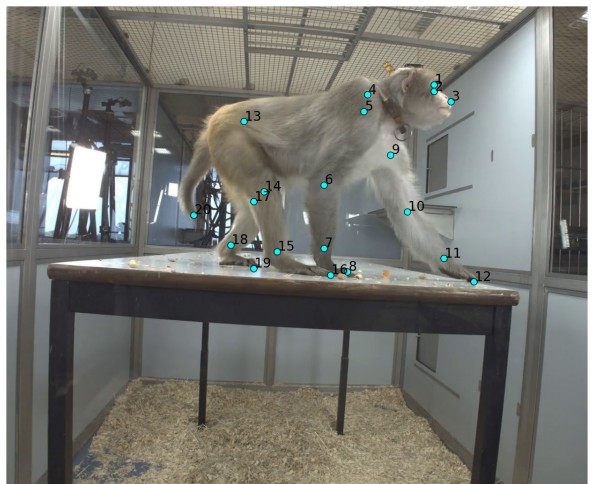

Figure 7: Predicted 2D keypoints of an exemplary frame.

## A.5 KEYPOINT ANNOTATION AND PROCESSING CONTROL STEPS

The 3D labeling procedure was carried out by two other annotators. To support keypoint annotation using our multi-camera calibration, once at least two views of a keypoint were labeled, we triangulated its 3D position using the direct linear transform (see also Appendix B.5) and projected the estimate into the remaining unlabeled views. In addition, for every annotated keypoint, epipolar lines were displayed in the remaining views, including cases with fewer than two annotated views, to guide the annotators toward geometrically consistent locations. This multi-view assistance substantially accelerated the annotation process. Annotators could refine existing labels or add additional keypoint annotations as needed. Before proceeding to the next body landmark, we required the mean reprojection error across all labeled views to remain below 5 pixels to ensure geometric consistency. Both annotators independently checked for keypoint swaps, after which the video acquisition specialist reviewed and refined the annotations when necessary. Each monkey body required 15 and 20 minutes of annotation time per annotator and an additional 10 minutes for review, totaling 140 hours of labeling effort for the 306 monkey bodies.

The reprojection errors across all 20 labeled monkey keypoints are shown in Table 8. These errors remain low, largely because the annotation interface supports zooming and produces geometrically consistent keypoints. To estimate the expected displacement between true and estimated 3D keypoints produced by the annotation tool, we sampled the entire table area and a volume up to 60 cm above it, resulting in 36 known 3D locations. After perturbing the 2D projections of these points, we computed 3D errors between triangulated (perturbed) annotations and their ground-truth positions. The estimated 3D locations remain highly accurate even when triangulated from only two annotated views under the reprojection threshold of 5 pixels (see Table 9). This scenario, or the more extreme case of 10 pixels, represents a conservative lower bound on annotation quality, since in practice also multiple viewpoints were annotated and all 2D keypoint projections from the 16 cameras were visible to the annotators.

Table 8: Reprojection error statistics in pixels (px) of the 20 body keypoints in the 3D Labeling Tool across the 306 labeled monkey bodies.

| Keypoint index | Keypoint name | Mean | Std |
|---|---|---|---|
| 1 | Right eye | 0.431 | 0.753 |
| 2 | Left eye | 0.341 | 0.502 |
| 3 | Nose | 0.369 | 0.500 |
| 4 | Neck | 0.980 | 1.252 |
| 5 | Right shoulder | 0.937 | 1.280 |
| 6 | Right elbow | 0.956 | 1.234 |
| 7 | Right wrist | 0.769 | 0.998 |
| 8 | Right hand tip | 0.330 | 0.531 |
| 9 | Left shoulder | 0.884 | 1.140 |
| 10 | Left elbow | 0.991 | 1.321 |
| 11 | Left wrist | 0.791 | 1.132 |
| 12 | Left hand tip | 0.277 | 0.460 |
| 13 | Hip | 1.027 | 1.419 |
| 14 | Right knee | 0.976 | 1.312 |
| 15 | Right ankle | 0.776 | 1.074 |
| 16 | Right foot tip | 0.327 | 0.544 |
| 17 | Left knee | 0.977 | 1.120 |
| 18 | Left ankle | 0.796 | 1.009 |
| 19 | Left foot tip | 0.320 | 0.523 |
| 20 | Tail | 0.707 | 1.086 |

Table 9: Mean $\pm$ std of the 3D Euclidean error (mm) of estimated 3D ground-truth points under different annotation noise levels and number of annotated cameras. Projected ground-truth points per annotation view were shifted randomly, and independently in $(x, y)$ image dimensions for $[-\sigma, 0, \sigma]$ pixels (px) before triangulation. Columns indicate the annotation noise $\sigma$; rows indicate the number of annotation cameras used for triangulation.

| # Cameras | $\sigma = 1$px | $\sigma = 2$px | $\sigma = 5$px | $\sigma = 10$px |
|---|---|---|---|---|
| 2 | $1.440 \pm 1.636$ | $2.990 \pm 5.374$ | $7.329 \pm 8.318$ | $15.506 \pm 31.656$ |
| 3 | $1.007 \pm 0.578$ | $2.011 \pm 1.219$ | $5.015 \pm 3.048$ | $10.045 \pm 5.768$ |
| 4 | $0.818 \pm 0.427$ | $1.638 \pm 0.841$ | $4.087 \pm 2.126$ | $8.202 \pm 4.216$ |

Bounding boxes were derived from the extremal or 2D keypoints closest to the image boundaries in each camera view. To ensure full spatial coverage of each monkey, we expanded the associated triangulated 3D keypoints by a safety margin of 10 cm along the plane orthogonal to the camera before projecting them back to 2D. This allowed us to retrieve bounding box labels from keypoint annotations.

We also applied a set of quality checks to labels generated by YOLOv8. Specifically, only individuals known to be present in the scene (according to our multi-view recording documentation) were considered, allowing us to filter out erroneously detected identities. For each individual, only the highest-confidence detection was retained. Missing or duplicate detections were flagged as erroneous, and the corresponding camera view was excluded for that frame in surface model fitting.

During mesh optimization, we additionally weight each camera viewpoint by its detection confidence. Furthermore, we also weight the contribution of individual keypoints using the confidence scores provided by HRNet-W48.

For the action recognition benchmark, we excluded actions for which less than 95% of timepoints could be linked to surface reconstructions for all individuals in the scene. This filtering is inherently scene- and video-dependent, since a monkey may only momentarily engage with another individual while otherwise remaining beneath the table, preventing the generation of labels and, consequently, surface reconstructions. In such cases, the optimization would fail, resulting in non-convergent pose reconstructions, i.e., frames in which the estimated pose parameters contain invalid (NaN) values.

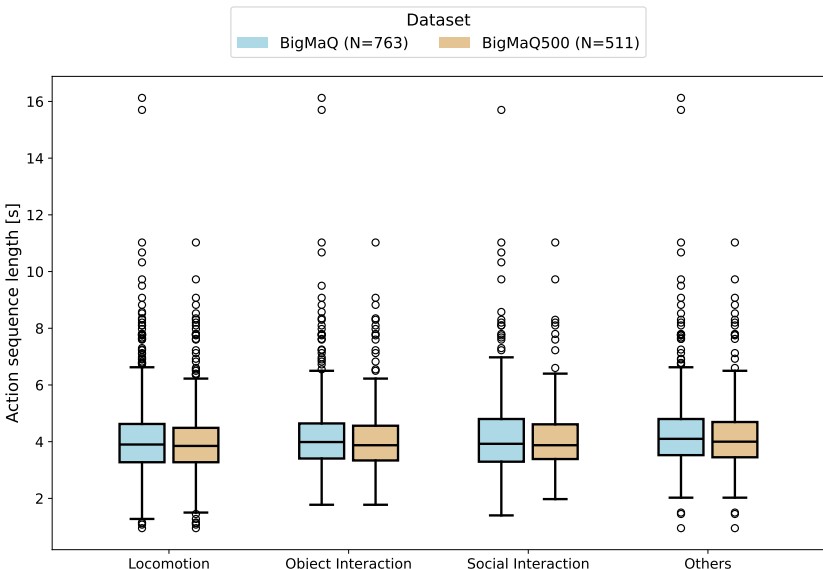

Figure 8: Box plots showing the distribution of action sequence lengths for `BigMaQ` and Big-MaQ500 across the four behavioral categories. The filtering step does not substantially affect the action length distribution in the action recognition benchmark.

## B    MESH OPTIMIZATION DETAILS

3D mesh tracking aims to recover the articulated motion and surface deformation of an individual across time using a parametric mesh driven by an underlying skeleton. Each mesh vertex is associated with a set of skinning weights $\mathbf{W}$ that specify how it moves under joint rotations $\boldsymbol{\theta}$; these weights are typically created during rigging of a high-poly template model (shown below) by an artist and can be automatically initialized and manually refined in commercial software such as Autodesk Maya or ZBrush. For computational efficiency, we derive a low-poly mesh from our high-poly template, preserving the same rig and skinning weights, which allows us to perform fast optimization while retaining the option to reconstruct higher-detail surfaces when needed (e.g., for neurophysiological stimulus creation). During tracking, subject-specific shape adaptation can op-

tionally be performed beforehand—using manual or high-accuracy 3D keypoint annotations together with silhouette information—to better fit individuals not well represented by an existing shape space, such as SMAL, after which the adapted model is used to fit full video sequences.

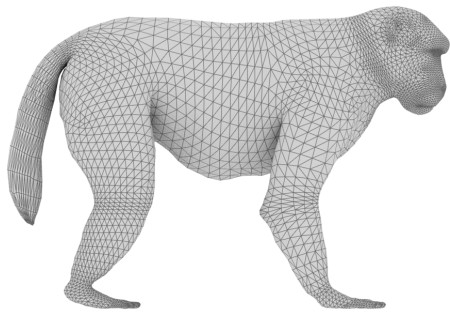

Figure 9: Side view of the high-poly mesh.

## B.1 ADAPTABLE MODEL PARAMETERS

Let us consider the extended version of equation 1 that includes parameters to adapt the vertices and joints for different individuals

$$\mathbf{V}_P = \gamma \cdot \mathbf{R} \cdot LBS(\boldsymbol{\theta}; \mathbf{V}(\boldsymbol{\xi}), \mathbf{J}(\boldsymbol{\alpha}), \mathbf{W}) + \mathbf{t}.$$

To adapt the mesh to different subject skeletal structures, we include learnable bone lengths $\boldsymbol{\alpha}$ in the function $\mathbf{J}(\boldsymbol{\alpha})$, shifting the initial joint locations $\mathbf{J}_R$ if needed as described in Badger et al. (2020); Wang et al. (2021b). The posed joints $\mathbf{J}_P$ are then determined by the relative body, and global pose transformations as with posed vertices $\mathbf{V}_P$. To account for differences in body shape beyond bone lengths, we introduce learnable offsets $\boldsymbol{\xi}$ for each vertex in the template mesh to yield shifted vertices denoted by

$$\mathbf{V}(\boldsymbol{\xi}) = \mathbf{V}_R + \mathbf{A} \cdot \boldsymbol{\xi}.$$

As the number of such offsets scales with the number of vertices in the mesh, we divide them into groups of symmetrical and unique vertices. This helps to further reduce the parameter space for finding subject-specific mesh offsets. We therefore include a sparse mapping $\mathbf{A} \in \mathbb{R}^{N_V \times 3795}$ with $\boldsymbol{\xi} \in \mathbb{R}^{3795 \times 3}$, where symmetrical vertices share weights, but the direction of the vector $\boldsymbol{\xi}_i$ is flipped. In the low-poly version, these offsets reduce to $\boldsymbol{\xi}_{LP} \in \mathbb{R}^{1305 \times 3}$. As we neither allow fingers to be posed in pose space nor their bone lengths to vary, we also exclude these vertices from $\boldsymbol{\xi}$, thereby further decreasing the number of learnable parameters. Consequently, only 27 joints participate in posture estimation. We denote this restricted number of joints for posing as $N_{J_R}$.

## B.2 LOSS FUNCTIONS

**Keypoint Reprojection Loss.** $L_{kp}^c$ defines the weighted mean squared error of 2D projected joint locations of the posed avatar $\mathbf{J}_P$, along with corresponding image keypoints $\mathbf{P}_c$ for a given camera $c$

$$L_{kp}^c(\Theta) = \frac{1}{\sum_{k=1}^{N_K} w_k} \sum_{k=1}^{N_K} w_k ||\Pi_c(\mathbf{J}_P^k) - \mathbf{P}_c^k||_2^2,$$

where $\Pi_c$ transforms the 3D points from global space into the camera's image plane. The weights $w_k$ manage the contribution of individual joints, where we emphasize the importance of the hip, and tail keypoints. In practice, we extend these fixed joint weights adaptively by the confidence scores provided by the 2D keypoint estimation model.

**Silhouette Reprojection Loss.** Let $\mathcal{S}$ denote the silhouette renderer and write the rendered mask as $\hat{\mathbf{S}}^{(c)} \in [0,1]^{I_H^c \times I_W^c}$ and the ground-truth mask as $\mathbf{S}^{(c)} \in \{0,1\}^{I_H^c \times I_W^c}$

$$\hat{\mathbf{S}}^{(c)} = \mathcal{S}(\Pi_c, \mathbf{V}_P, \mathbf{F}).$$

The per-camera silhouette loss normalizes by image size and is accumulated only for views with sufficiently small keypoint reprojection error

$$L_{\text{sil}}^c(\Theta) = \begin{cases} \dfrac{1}{I_W^c I_H^c} \left\| \hat{\mathbf{S}}^{(c)} - \mathbf{S}^{(c)} \right\|_2^2, & \text{if } L_{kp}^c(\mathbf{J}_P) < \sigma_{kp}, \\ 0, & \text{otherwise.} \end{cases}$$

This follows Rüegg et al. (2022) with the addition of the dimension-based normalization to avoid resolution-dependent scaling.

**Pose and Bone Constraints.** We penalize poses that strongly deviate from the template four-legged standing pose to favor more realistic poses using the error term

$$L_P(\boldsymbol{\theta}) = \frac{1}{\sum_{k=1}^{N_{J_R}} w_\theta} \sum_{k=1}^{N_{J_R}} w_\theta \|\boldsymbol{\theta}_k\|_2.$$

This suppresses uncontrolled joint rotations of the reduced set of $N_{J_R}$ joints of the kinematic tree, such as those of the hand, which are not sufficiently constrained by the set of keypoints. Similar to the keypoint reprojection loss $L_{kp}^c$, we also impose per-joint weights to specifically allow more flexibility in tail joints.

As in Badger et al. (2020), we penalize individual bone lengths outside pre-defined limits $(\alpha_{\min}, \alpha_{\max})$ by $L_b(\alpha_i) = \max(0, \alpha_i - \alpha_{\max}) + \max(0, \alpha_{\min} - \alpha_i)$ for individual bones. We calculate the entire bone length loss as the sum of the squared individual error terms $L_b(\boldsymbol{\alpha}) = \sum_i L_b(\alpha_i)^2$.

**Shape Refinement.** We further adapt the surface shape of the model for individual differences apart from bone lengths $\boldsymbol{\alpha}$. Therefore, we define a smoothness loss of neighboring vertex offsets of the entire mesh by

$$L_{sm}(\boldsymbol{\xi}) = \sum_{i=1}^{N_V} \sum_{v_j \in \mathcal{N}(v_i)} \|(\mathbf{A}\boldsymbol{\xi})_{i,:} - (\mathbf{A}\boldsymbol{\xi})_{j,:}\|_2^2,$$

where $v_j$ is a vertex within the neighborhood $\mathcal{N}$ of $v_i$. With $()_{i,:}$, we denote the i-th row of a matrix, so that $(\mathbf{A}\boldsymbol{\xi})_{i,:}$ and $(\mathbf{A}\boldsymbol{\xi})_{j,:}$ represent the respectively learned $i$-th and $j$-th vertex offsets.

## B.3 ANGULAR VELOCITY

Let $\boldsymbol{\theta}_j^{(n)} \in \mathbb{R}^3$ denote the axis–angle vector of joint $j$ at time step $n$, with

$$\phi_j^{(n)} = \|\boldsymbol{\theta}_j^{(n)}\|_2, \qquad \mathbf{u}_j^{(n)} = \frac{\boldsymbol{\theta}_j^{(n)}}{\|\boldsymbol{\theta}_j^{(n)}\|_2}.$$

The per–joint angular velocity (Bauchau & Trainelli, 2003) is approximated by finite differences as

$$\boldsymbol{\omega}_j^{(n)} \approx \frac{1}{\Delta t} \left( \Delta\phi_j^{(n)} \mathbf{u}_j^{(n)} + \left[ \sin\phi_j^{(n)} \boldsymbol{I} + \left(1 - \cos\phi_j^{(n)}\right)[\mathbf{u}_j^{(n)}]_\times \right] \Delta\mathbf{u}_j^{(n)} \right),$$

where

$$\Delta\phi_j^{(n)} = \phi_j^{(n+1)} - \phi_j^{(n)}, \qquad \Delta\mathbf{u}_j^{(n)} = \mathbf{u}_j^{(n+1)} - \mathbf{u}_j^{(n)},$$

and $[\mathbf{u}]_\times$ denotes the $3 \times 3$ skew–symmetric matrix.

### B.4 OPTIMIZATION SCHEME

To optimize the articulate mesh model, we use Adam (Kingma & Ba, 2017) with adaptive learning rates on epochs and parameters. Before optimizing $\Theta$, we initialize $\mathbf{R}, \mathbf{t}$ by a Procrustes transformation to speed up the alignment and prevent weird joint rotations that can occur when simultaneously optimizing $\mathbf{R}$ and $\boldsymbol{\theta}$.

**Individual Shape Adaptation.** To create for each monkey a subject-specific mesh model, we adapt model parameters in a stage-wise manner. First we align the poses to match the manual label data. Then, we adapt the mesh for multiple views and keyframes of the same monkey, followed by coloring the resulting surface. An overview of the resulting meshes is given in Figure 10, and the ranges of the adapted parameters across individuals is further quantified in Table 10. Afterwards, we use these surface models to derive the monkey's individual behavior over time. Individual loss weights $\lambda_i$ are given below.

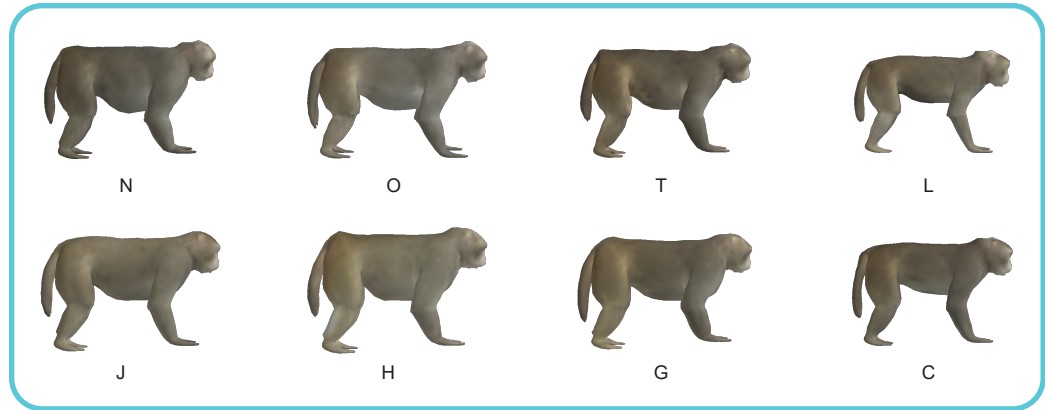

Figure 10: An overview of the individual mesh adaptations for all eight monkeys in `BigMaQ`.

Table 10: Summary statistics for adapted parameters in individual shape adaptation (min, max, mean, std).

| Parameter | Min | Max | Mean | Std |
|---|---|---|---|---|
| Global Scale $\gamma$ | 0.791 | 1.001 | 0.908 | 0.060 |
| Bone Lengths $\boldsymbol{\alpha}$ | 0.833 | 1.179 | 0.973 | 0.124 |
| Vertex Offsets $\boldsymbol{\xi}$ | -0.030 | 0.039 | 0.000 | 0.005 |
| Vertex Colors $\mathbf{C}$ | 26.854 | 254.747 | 164.602 | 35.398 |

**Dynamic Pose Fitting.** We detect the cameras to be used in optimization either by switching a minimal set of cameras in a greedy manner, or by a random selection of available cameras per frame. The former would select the cameras for the longest non-interrupted sequence of frames, and switch only to another camera if further labels can not be provided. This setting was used to generate the poses in `BigMaQ`, which increases temporal smoothness compared to a random selection. For comparisons with other methods, we selected actions that provide a continuous stream of six cameras, since dynamic camera changes are not supported in MAMMAL.

Table 11: A typical set of hyper-parameters for shape and dynamic pose estimation used for the `BigMaQ` dataset. The first three columns are involved in individual shape fitting, whereas the last column refers to the stage of dynamic pose fitting.

|              | Pose | Mesh | Color | Time |
|--------------|------|------|-------|------|
| $\lambda_P$    | 3    | 0    | 0     | 3    |
| $\lambda_b$    | 100  | 100  | 0     | 0    |
| $\lambda_{sm}$ | 0    | 5    | 0     | 0    |
| $\lambda_{kp}$ | 2    | 1    | 0     | 1    |
| $\lambda_{sil}$| 1500 | 100  | 0     | 50   |
| $\lambda_C$    | 0    | 0    | 1     | 0    |
| $\lambda_T$    | 0    | 0    | 0     | 10   |
| $\sigma_{\mathrm{kp}}$ | 10 | — | — | 5 |
| Epochs       | 1200 | 150  | 30    | 350  |

## B.5 3D ERROR METRICS

To quantify the alignment of the tracked surface with the keypoints, we measure the mean per-joint position error (MPJPE) between the triangulated 3D keypoints $\mathbf{P}$ and the corresponding posed joint locations $\mathbf{J}_P$

$$\mathrm{MPJPE}(\mathbf{J_P}, \mathbf{P}) = \frac{1}{N_K} \sum_{k=1}^{N_K} \|\mathbf{J}_P^k - \mathbf{P}^k\|_2.$$

When reporting MPJPE for an entire action sequence, we compute the error per frame and then average these values across time.

To evaluate temporal smoothness of the posed keypoints over an action of length $T$, we follow Li et al. (2023) and use the mean per-joint temporal deviation (MPJTD)

$$\mathrm{MPJTD}(\mathbf{J}_P) = \frac{1}{T-1} \frac{1}{N_K} \sum_{n=1}^{T-1} \sum_{k=1}^{N_K} \|\mathbf{J}_P^{(n,k)} - \mathbf{J}_P^{(n+1,k)}\|_2,$$

which corresponds to the mean velocity of posed joints across time.

We triangulate keypoints using the direct linear transform with random sample consensus (RANSAC) following Karashchuk et al. (2021). For each marker and timestep in the predicted keypoint trajectories, we randomly select up to five camera views to generate 3D point proposals. Among these proposals, we retain the point $\mathbf{P}$ with the lowest mean reprojection error across the selected cameras. When triangulating annotator-provided keypoints in the 3D labeling tool, we use all available annotations due to the substantially lower computational cost.

## B.6 ABLATIONS

For dynamic pose estimation, we ablate different $\lambda$'s to give an intuition about their influence in the optimization procedure of an entire action of 123 frames or $\sim 3$ s in Figure 11 and the associated Table 12. Keypoints are an integral part of the optimization procedure, and without the model remains relatively fixed (see MPJTD value below) at wrong locations. More specifically, the silhouettes are not sufficient to align the mesh with multi-camera data. However, the silhouettes do have an impact on reconstruction quality. This is apparent since without temporal loss the IoU scores remain relatively high even though the extremities can end up in an unnatural sequence of joint rotations. The $\lambda_P$ pose prior is particularly helpful for preventing large pose deviations from the template during static shape estimation. In dynamic pose estimation, removing this term does not affect the quality metrics for this example. Nevertheless, we generally retain it due to its negligible computational

cost and the additional robustness it provides, even though the temporal alignment already fulfills a similar role in correcting rotational artifacts.

We further evaluate whether incorporating bounding-box and keypoint confidence values improves surface fitting, and how the number of available viewpoints affects dynamic pose reconstruction, as summarized in Table 13. Excluding confidence values consistently degrades surface fitting quality across all metrics. Likewise, reducing the number of viewpoints lowers accuracy in every measure, with the most pronounced impact observed in the mean per-joint position errors.

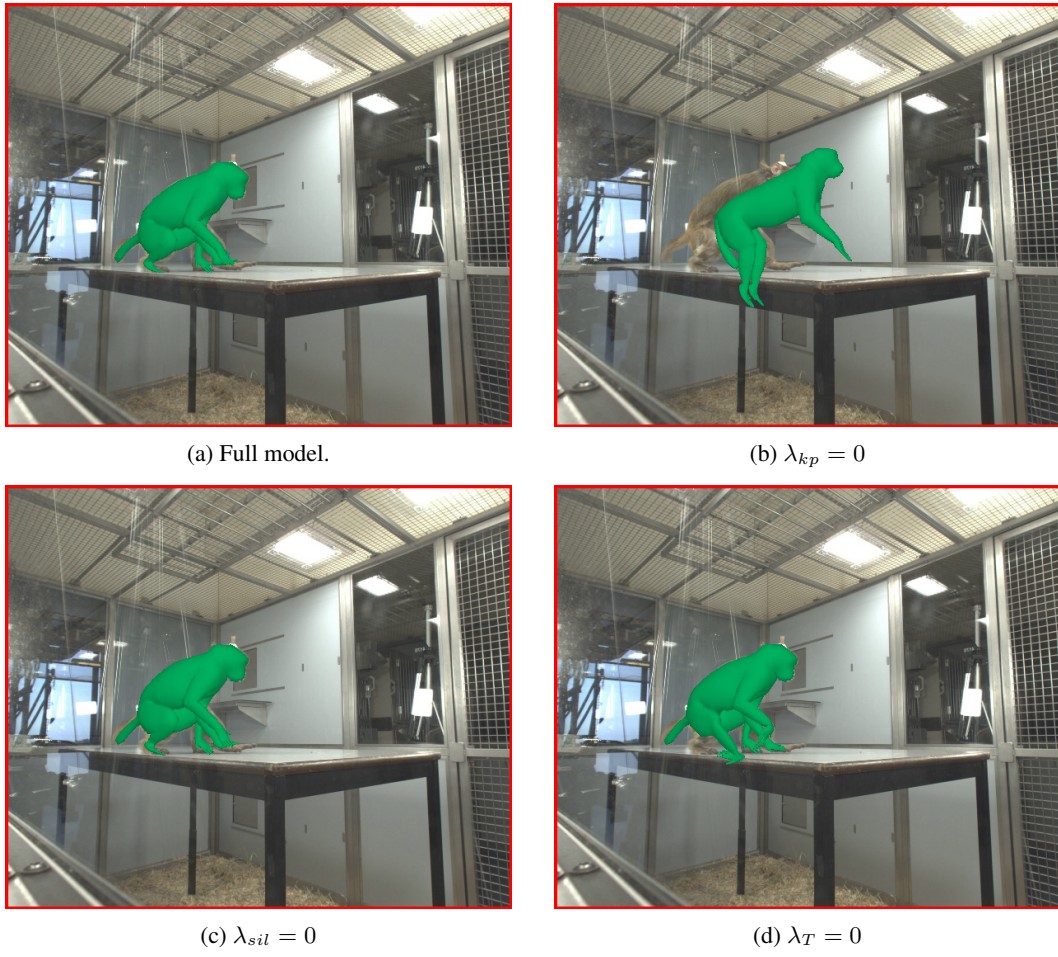

(a) Full model.

(b) $\lambda_{kp} = 0$

(c) $\lambda_{sil} = 0$

(d) $\lambda_T = 0$

Figure 11: Surface fitting results when disabling relevant $\lambda$ entries in dynamic pose estimation. The red outline indicates that this particular camera was not involved in surface fitting, ensuring an unbiased different viewpoint to evaluate reconstruction quality.

Table 12: Ablation experiment on weighting factors $\lambda$ in dynamic optimization: mean error metrics for the entire time sequence of the action shown in Figure 11. IoU scores are averaged over time and across cameras involved in the surface reconstruction. Arrows indicate whether lower or higher values are better.

| Ablation Setting | MPJPE $\downarrow$ (mm) | MPJTD $\downarrow$ (mm/frame) | IoU $\uparrow$ |
| --- | --- | --- | --- |
| None (full model) | 16.479 | 9.076 | 0.855 |
| $\lambda_{kp} = 0.$ | 210.081 | 1.844 | 0.375 |
| $\lambda_{sil} = 0.$ | 16.593 | 9.344 | 0.812 |
| $\lambda_T = 0.$ | 17.082 | 10.620 | 0.853 |

Table 13: Ablation experiment on confidence-value integration and viewpoint reduction in dynamic optimization: mean error metrics for the entire time sequence of the action shown in Figure 11. IoU scores are averaged over time and across cameras of the full model ($N_{\text{views}} = 6$). Arrows indicate whether lower or higher values are better.

| Ablation Setting | MPJPE $\downarrow$ (mm) | MPJTD $\downarrow$ (mm/frame) | IoU $\uparrow$ |
|---|---|---|---|
| None (full model) | 16.479 | 9.076 | 0.855 |
| w/o conf. values | 18.676 | 9.756 | 0.847 |
| $N_{\text{views}} = 5$. | 19.052 | 9.971 | 0.848 |
| $N_{\text{views}} = 4$. | 29.322 | 11.085 | 0.822 |
| $N_{\text{views}} = 3$. | 26.286 | 11.138 | 0.825 |
| $N_{\text{views}} = 2$. | 36.053 | 11.902 | 0.801 |

## B.7 FAILURE CASES

If the mesh optimization fails, it is usually due to the quality of the labels used in alignment. While the SAM 2 silhouettes are generally of good quality (see also the Supplementary Video), some frames can contain silhouettes that include background features, combined or partial individuals. For this reason, we reduced the silhouette contribution in the time-optimization stage (see Table 11). Incorrect detections, and consequently incorrect keypoint predictions, can degrade the fitting quality, as illustrated in Figure 12 (first row, third column). Although such errors would need to persist across viewpoints and over multiple frames due to the temporal consistency constraint, establishing detection agreement across views during camera selection could help mitigate these issues.

In an otherwise well-tracked sequence of an individual, conflicting input signals for a particular frame can cause the temporal loss to lock in an incorrect orientation, especially when strong emphasis is placed on extremities, hands, and feet (second row). This failure case may be mitigated by an initial per-frame optimization without temporal constraints, or by increasing the weight of the pose prior.

When animals leave the capture volume or make contact with the glass, the estimated keypoints can become distorted or squashed, in part because such cases are underrepresented in the pose estimator's training data (third row). Unlike the previous cases, this type of failure is tied to the capturing setup and cannot be easily resolved without altering the enclosure or recording conditions.

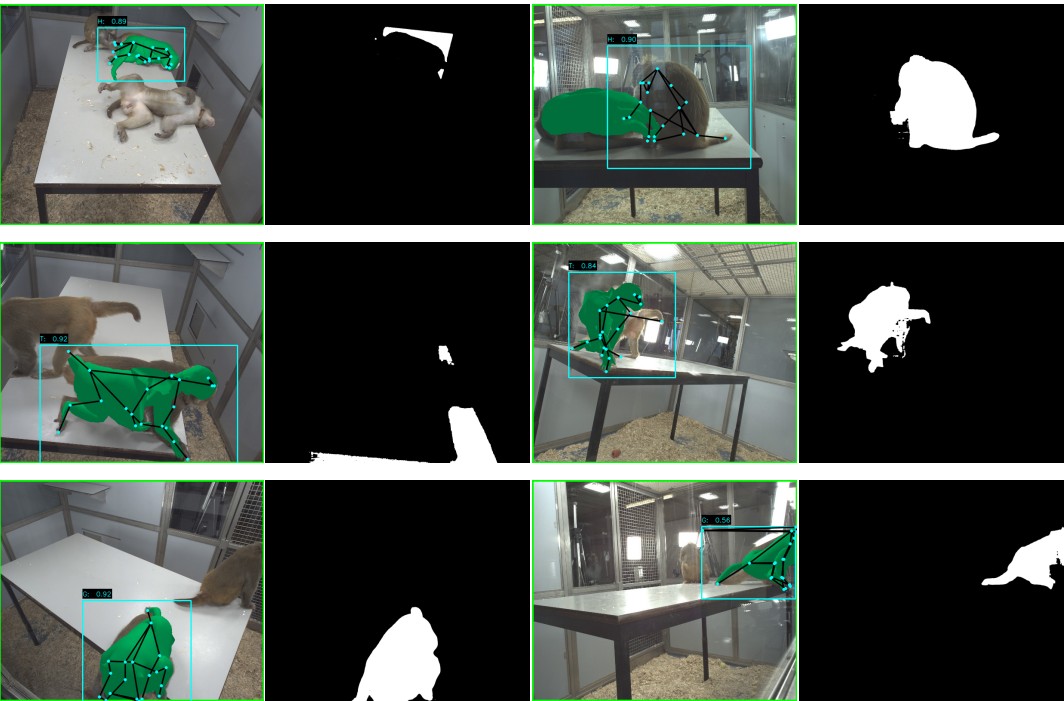

Figure 12: Failure cases of a frame in an action. The first and third column shows the rendered mesh on top of an image of the particular scene, including bounding box and keypoints labels, where the green frame outline indicates the camera's participation in optimization. The associated SAM 2 silhouette is shown in the second and fourth column, respectively. Rows show results for different individuals.

## C   ACTION RECOGNITION DETAILS

**Implementation.**   All sequences were padded to the maximum action length, with padding masks provided to a transformer consisting of 2 layers, each with 8 heads and a model dimension of 256. Appearance and pose features are concatenated and fused through a linear layer before being passed to the transformer. Each modality is first encoded by a two-layer multilayer perceptron (MLP), following (Rajasegaran et al., 2023). For models using spatial embeddings (patch tokens), an additional multi-head attention module is applied in the visual stream before the two-layer MLP. For the pose-only and visual-only baselines, the disabled modality is replaced with a zero-feature vector, ensuring that the model processes only a single stream while keeping the overall architecture unchanged. All models were trained for 20 epochs with a 70%/10%/20% stratified train/validation/test split on actions using AdamW (Loshchilov & Hutter, 2019) and EarlyStopping. All individuals and action categories are present in all splits. Viewpoints of the same action are not shared across splits. For training runs solely based on 3D information, where the same input would be provided for multiple camera views, we report 5-fold cross validation results. In cases where visual information is used, we report mean and standard deviation results of three independent training runs. For pose-based training, $\theta$ is mapped to rotation matrices.

## D   COMPUTATIONAL REQUIREMENTS

Action tracking, label generation, and surface fitting was executed on a single NVIDIA Geforce RTX 3090 GPU. A single optimization step in surface fitting takes 0.019 s, resulting in an effective per-frame runtime of 6.65 s for our default 350-epoch setting.

Inference of a single video for behavior recognition requires from 525.82–729.05 M FLOPs, assuming pre-computed visual and pose features. Of this, the pose stream accounts for approximately 30 M FLOPs. Training the feature-processing heads for the full 20 epochs at 10 Hz (for both the visual

and pose streams) takes 3-4 minutes for ViT-base-cls and up to 120 minutes for VideoPrism-base features on the BigMaQ500 data split.

