# OpenReview forum: "BigMaQ: A Big Macaque Motion and Animation Dataset Bridging Image and 3D Pose Representations"
_ICLR.cc/2026/Conference — ICLR 2026 Poster_

### Official Review · Reviewer_6cL8 · 2025-10-28

**Soundness:** 2
**Presentation:** 2
**Contribution:** 4
**Rating:** 4
**Confidence:** 3

**Summary:**

The authors present BigMac3D, a large dataset derived from hundreds of scenes of interacting rhesus macaques in a laboratory setting. Each scene is accompanied by 3D pose descriptions in the form of keypoints and textured surface meshes; per-view 2D keypoints and segmentation masks; and per-scene action labels. The authors then provide baseline action recognition results using a variety of pose and video descriptors, and demonstrate model improvements when high-quality pose descriptors are added to video descriptors.

**Strengths:**

BigMac3D presents a highly valuable resource to the behavioral quantification community. The dataset is very complete in terms of videos, keypoints, meshes, masks, etc., as well as the behavior annotations. This allows various types of methods developers to utilize this dataset.

The authors have also provided action recognition performance for a variety of baseline models, which is useful for methods developers looking to benchmark their models on this dataset.

**Weaknesses:**

The manuscript as written lacks clarity in various ways. The distinction between BigMac3D and BigMac500 isn't obvious. The types and amount of data related to each dataset is also not clear. The actual pipeline for collecting the raw data and creating the derived quantities (3D labels, meshes, etc.) is never clearly laid out. Addressing these issues would vastly improve the mansuscript.

The manuscript also lacks enough details about the quality of the dataset. The metrics in Tables 5+6 are a start but these numbers should, for example, be shared by behavior category. Example videos overlaid with keypoints and/or bounding boxes and/or segmentation masks would provide better intution for dataset quality. It is common to perform some kind of post-processing on the outputs of various models (detectors/pose estimators/segmenters) to clean up obvious errors; either this was not done or it was not described. These sorts of quality control steps are crucial before releasing such a large dataset to the community.

**Questions:**

Section A.4 describes the performance of the detection and pose estimation pipelines, but it is not clear from the reported metrics how less-than-perfect detections/poses influence mesh optimization. Furthermore, it is conceivable that these numbers will be highly dependent on the behavioral category, so it might be helpful to break them down this way. Section B.5 addresses this briefly, but I feel there should be much more space allocated to these issues.

Are there any post-processing steps used for the detection and pose estimation steps? For exmaple, multi-view (in)consistency across views could help flag detection errors.

There is no quantification of the segmentation masks from SAM2. Have the authors verified the performance of this model in any way? Has any post-processing been applied to clean up the segmentation masks? Presumably the predicted silhouettes would be helpful here? Is there a reason to include the SAM2 segmentation masks and not just the silhouettes output by the mesh-tracking stage?

It is not clear what the high-poly template model is used for.

How are skinning weights W chosen?

I am not super familiar with 3D mesh-tracking; it would be useful for that section to start with a few-sentence overview of the process before diving into specifics. For example, the appendix mentions this briefly, but it's not clear that the first step is to perform individual shape adaptation using the manual annotations, and then this information is used for the videos - this is not clear in the main text. Some other confusions I have:
- low- vs high-poly meshes, when are they used/not used
- SAM2 masks - are these used at all in the optimization process? If not, are they ever used to validate pieces of the optimization process?
- when are ground-truth keypoint annotations used in the pipeline, versus keypoints predicted by the pose estimation model?

The loss function in Eq. 2 contains a large number of hyperparameters. These values are given in Table 7 but there are no details on how these are chosen, or how sensitive the different hyperparameter values are.

Table 2: it is unclear what this actual metric is. The text says "IoU of mesh fits over time" - what is "ground truth" here? Is it IoU of meshes between time t and t+1? Table 2 legend says "first frame of the respective action", which seems contradictory. Please clarify.

L377: how is "reliable pose reconstruction" defined?

L411: the Kolesnikov reference should be Dosovitskiy

The conclusions from the action recognition results section are't entirely fair. The pose information is (presumably) sampled at 40 Hz, while the video information is sampled at 10 Hz. This confounds any direct comparison between the two. The more proper way to conduct this experiment to make the claims the authors want would be to downsample to pose data to 10 Hz or upsample the video data to 40 Hz. It is true that the former will lead to overall lower performance, and the latter to increased compute, and the hybrid results presented in the current submission could provide a useful approach for trading off accuracy and compute. But one of the aforementioned control experiments should be done.

Regarding the effects of pose representation, Blau et al. (https://arxiv.org/abs/2407.16727) looked at the incorporation of time into the pose representation and found a large performance improvement across multiple datasets. BigMac3D cool dataset to test these findings across a wider variety of spatial representations, if they are so inclined.

It's not actually clear what the difference between BigMac3D and BigMac500 are - BigMac3D has 750 "scenes", while BigMac500 has 8k "videos". Is one a subset of the other? Does BigMac500 not have segmentation masks, 2D keypoints, etc.? In general, a table with more exact details about what data is included in each dataset would help clarify this confusion.

Do the authors plan to publicly release the manual keypoint annotations? These could be very valuable for developers of 3D/multi-view pose estimation models.

It would be useful to provide train/test split information for the action recognition part so that new users can compare other models.

---

> ### Author Response · Authors · 2025-11-20
>
> We thank the reviewer for the comprehensive review, and we appreciate the positive assessment of our work. Below, we elaborate on the changes incorporated into the revised manuscript and address your questions. Because of rebuttal character limits, we quote your comments in abbreviated form below.
>
> > The manuscript as... would vastly improve ...
>
> We now include a new Figure 3 in the main text, which clarifies how the pipeline components relate to the individual datasets.
>
> > The manuscript also ... category.
>
> The 3D pose generation process with our 3D labeling tool was performed in parallel to behavior classification and was designed for pose variety with a balanced distribution of individuals in the training data. For this reason, we did not split the performance by behavioral category. However, the performance in the action recognition task already indicates the difficulties in social action recognition, consistent with previous findings for visual-only models. We have extended the information formerly in Tables 5 and 6 with additional intuition about the detector performance in Figure 5 that queries SAM 2 for masks and crops images for HRNet-W48.
>
> >  ... Example videos ... community.
>
> We now describe our post-processing steps and confidence-based optimization in Appendix A5. Side-by-side examples of labels and surface reconstructions will be included in the Supplementary Video.
>
> > Section A.4 ... mesh optimization.
>
> To address this, we now include a loss ablation in Figure 9 and Table 10. Because keypoints are critical for accuracy, we describe processing controls now in Appendix A5.
>
> > There ... masks from SAM2. ... mesh-tracking stage?
>
> SAM 2 is used as a zero-shot foundation model to provide temporally consistent masks for aligning our mesh models with video frames, as now clarified in Figure 3. Although projected silhouettes from the mesh could be used in downstream tasks, SAM 2 masks ensure reproducibility of the 3D reconstruction pipeline itself. During optimization, silhouettes are also weighted by bounding-box confidence, but overall they are less critical to dynamic pose accuracy than keypoints.
>
> > It is ... used for.
> > How ... chosen?
> >... 3D mesh-tracking; it would be useful ... the main text.
>
> Thank you for this suggestion. We now added an introduction to the topic to Appendix B that is referenced in the main text. Shape adaptation is not required for mesh optimization but we highly recommend it for species not represented by a parametric shape space such as SMAL. In fact, our work is inspired by previous works on static images in birds, and we were intrigued by the benefits of applying this idea to video data. We hope that reconstruction gains are apparent in our comparisons with MAMMAL. Because keypoints are critical for accurate surface reconstruction, we leveraged accurate 3D keypoints from our labeling tool to estimate subject-specific deformations, rather than relying on noisier 2D predictions.
>
> >The loss function in Eq. 2 ... values are.
>
> We now include qualitative and quantitative analyses of hyperparameter effects in Figure 9 and Table 10. The parameters are species-dependent and typically adjusted manually. Some configurations yielded correct hand and foot rotations only after introducing additional keypoints, highlighting their importance. To tune the weights, we monitored individual loss terms across optimization iterations for exemplary actions. We will release this visualization tool with the code.
>
> > Table 2: it is ... Please clarify.
>
> IoU is computed between SAM 2 masks and projected surface fits across viewpoints. For multi-view surface-tracking approaches (BigMac3D, MAMMAL), we compute IoU across both viewpoints and timepoints for the actions shown in Figure 4 (formerly Figure 3). Note that we have now extended this evaluation with two 3D keypoint alignment metrics.
>
> > L377 ... ?
>
> Now specified in Appendix A5.
>
> > L411
>
> Corrected.
>
> > The conclusions from ... done.
>
> We thank the reviewer for noting this. We now clarify that **videos and surface reconstructions were both sampled at 10 Hz**, ensuring a fair comparison across data streams.
>
> > Regarding the effects ... inclined.
>
> Thank you for bringing this to our attention. We agree that a more detailed investigation of temporal encoding in pose representations beyond our architecture is an interesting direction for future work. We also believe that BigMac3D provides a suitable testbed for evaluating such effects across a wide range of 3D representations, including whether different modalities can be effectively compressed.
>
> >Do the authors plan to ...
> >... compare other models.
>
> Yes. We will release the manual keypoint annotations, the 3D labeling tool, and the train/test splits for the action-recognition benchmark upon publication.
>
> We thank the reviewer again for the detailed suggestions to improve clarity and accessibility. With the incorporated changes, we believe the manuscript has been substantially strengthened.

---

> > ### Comment · Reviewer_6cL8 · 2025-11-22
> >
> > I thank the authors for their response: Figure 3, Appendix A.5, and the intro to Appendix B (among other updates) have been helpful additions. I look forward to seeing the supplementary video.
> >
> > Another question about the final dataset: will camera calibration information be released for the various videos?
> >
> > For pose estimation and detection models, it appears that individuals appear in train, val _and_ test splits (please correct me if I'm wrong). This makes the high values in Tables 6 and 7 perhaps a bit less surprising. I suppose these models are not intended to generalize well, they only need to work within the set of individuals in the benchmark, but the details of the training splits should be more explicit.

---

> > > ### Author Response · Authors · 2025-11-30
> > >
> > > We want to thank the reviewer for the follow-up response.
> > >
> > > We have now updated the Supplementary Video to show the labels used for alignment together with the raw videos and the surface reconstructions side by side.
> > >
> > > Yes, we will include the full camera calibration data (geometric and color) for all recording sessions.
> > >
> > > Your observation is correct, and at this stage the model is intended to operate within our recording sessions. We have made the details of the training splits more explicit in the main text (Section 4, Action Recognition), which now reads:  _“Data splits used to train these models included all individuals and action categories. More information on the model architecture and training is provided in Appendix C.”_
> > >
> > > Thank you again for your participation in the discussion and for the positive assessment of the changes we incorporated.

---

### Official Review · Reviewer_Q757 · 2025-10-30

**Soundness:** 3
**Presentation:** 4
**Contribution:** 4
**Rating:** 8
**Confidence:** 3

**Summary:**

This paper introduces BigMac3D, a large-scale dataset of over 750 multi-view video scenes capturing rhesus macaques performing diverse individual and social behaviors. Using 16 synchronized calibrated cameras, the authors reconstruct 3D surface-based pose and shape representations through a markerless motion-capture pipeline with skeletal rigs (115 joints) and subject-specific meshes. The dataset provides pose vectors, segmentation masks, keypoints, bounding boxes, and ethogram-aligned action labels, forming the first large-scale benchmark coupling 3D surface pose with action recognition in non-human primates. A derived benchmark, BigMac500, evaluates how these pose descriptors improve action classification performance when fused with various vision backbones (ResNet50, ViT, DINOv2, VideoPrism), achieving consistent mAP gains (≈+8–12%)

**Strengths:**

1.The dataset is unprecedented in scale and annotation richness for macaques, combining high-fidelity 3D surface tracking with ethogram-based behavioral labels across multi-individual interactions

2.The work builds on recent multi-view surface tracking advances (MAMMAL, AniMer+) but contributes practical improvements, such astemporal regularization, cropped differential rendering, and texture fitting, to enable efficient processing of thousands of sequences.

3.Comparisons against MAMMAL and AniMer+ confirm clear reconstruction gains (IoU = 0.84 vs 0.71/0.59). The action-recognition experiments further demonstrate that incorporating 3D pose vectors significantly boosts mAP across all tested encoders.

4.The dataset addresses a genuine need in neuroscience and ethology by bridging vision models with interpretable 3D motion cues, promoting reproducible behavioral analyses for NHP research.

**Weaknesses:**

1.The paper omits several critical details necessary for reproducibility, such as frame selection criteria, train/test split definitions, and subject-level separation (e.g., whether BigMac500 avoids identity leakage across splits). Although the authors describe a 70/10/20 split for the BigMac500 action recognition subset, this rule applies only to that benchmark, not the full BigMac3D dataset. There is no explicit statement on whether subjects are disjoint across splits. Furthermore, while the dataset uses a 16-camera synchronized capture setup, no visualization of camera IDs, baseline distances, or view diversity is provided in any figure or table, making it difficult to assess spatial coverage or view redundancy. These omissions weaken the dataset’s clarity and reproducibility.

2.While the authors claim “high-quality surface reconstruction,” the paper provides no quantitative 3D validation, such as per-joint reconstruction error or ground-truth comparison. Qualitative figures (e.g., Fig. 3) indeed show visually strong fits, but the absence of uncertainty estimates, human-verified subsets, or error histograms leaves the actual accuracy unquantified. This weakens confidence in downstream claims about 3D pose reliability and consistency across subjects.

3.Related to the first issue, the lack of clarity on subject-level isolation across training and testing sets raises potential concerns about identity memorization in the reported benchmarks. Explicitly ensuring disjoint subjects would help establish a fair evaluation protocol for behavior recognition tasks.

**Questions:**

Could the authors provide more details on how the dataset is organized and validated? In particular, it would be helpful to clarify how the training and testing splits are defined for BigMac3D and BigMac500, and whether subjects are disjoint across splits to avoid identity memorization. Additionally, more explanation on how 3D pose accuracy and annotation confidence were assessed would strengthen the paper’s reliability. Finally, reporting key dataset statistics such as frame sampling rules, camera ID distributions, and sequence lengths would improve the dataset’s transparency and reproducibility.

---

> ### Author Response · Authors · 2025-11-20
>
> Thank you for your thorough review; we appreciate your confidence in the value and broader impact of our work. We would like to respond at this stage to some of your suggestions, given the changes that we have incorporated in the manuscript:
>
> > Although the authors describe a 70/10/20 split for the BigMac500 action recognition subset, this rule applies only to that benchmark, not the full BigMac3D dataset.
>
> BigMac3D provides action labels and dynamic surface reconstructions for all actions that can be publicly released. From these, we include only successful surface reconstructions for all individuals into the action benchmark BigMac500, as now described in Appendix A5. The new Figure 3 further addresses the clarity and dependencies within the pipeline and corresponding data. In addition, the dataset for training the pose and detection models is now described in more detail in Appendix A4, along with further quality control steps in Appendix A5. Given the strong influence of keypoints on surface estimation quality, as demonstrated in the ablation in Figure 11, our pipeline included uncertainty estimates per camera as bounding box and individual keypoint confidences provided by YOLOv8-L and HRNet-W48.
>
> As in the longitudinal behavioral dataset ChimpACT (Ma et al., 2023), we ensured that all identities and action categories are present in the benchmark splits.  We added this information to Appendix C. Note that in contrast to Rajasegaran et al. (2023), we do not incorporate body-shape parameters into the pose stream, which could reveal individual identity beyond what visual features may already encode. For your reference, an overview of these subject-specific parameters is now given in Figure 10 and Table 10.
>
> > 2.While the authors claim “high-quality surface reconstruction,” the paper provides no quantitative 3D validation, such as per-joint reconstruction error or ground-truth comparison.
>
> Regarding quantitative 3D validation, the IoU values are computed across multiple viewpoints and are therefore complementary to true 3D metrics. We now also provide two additional measures that evaluate the skeletal alignment of our surface models in comparison to MAMMAL for single-subject actions: a) the mean per-joint position error (MPJPE, in mm) and b) the temporal smoothness of these joints measured by the mean per-joint temporal deviation (MPJTD, mm/frame), which increases for noisier skeleton trajectories. In both measures, we retain the substantial improvements over MAMMAL.
>
> We thank the reviewer again for their supportive assessment of our work. Kindly note that we will provide the additional requested statistics (camera-ID and sequence-length distributions) in our next comment. We believe that these additions will further strengthen the clarity and completeness of the dataset.

---

> > ### Author Response · Authors · 2025-11-30
> >
> > We are sorry that the discussion ended prematurely. Below, we provide the remaining information regarding camera-ID and sequence-length distributions.
> >
> > We have added Figure 5 in the Appendix to show the spatial distribution of camera IDs around the enclosure. We have also included Figure 8, which presents the distribution of action lengths for BigMac3D and BigMac500.
> >
> > Thank you again and we appreciate your acknowledgment of this work’s relevance to behavioral and neuroscience research in non-human primates.

---

### Official Review · Reviewer_Maxm · 2025-10-31

**Soundness:** 3
**Presentation:** 4
**Contribution:** 3
**Rating:** 2
**Confidence:** 4

**Summary:**

The paper presents BigMac3D, a large-scale 3D dataset of macaques that bridges visual appearance and 3D pose representations for action recognition. The dataset comprises over 750 multi-view scenes recorded with 16 synchronized cameras and annotated with 3D skeletal joint rotations, segmentation masks, 2D keypoints, and ethogram-aligned behavioral labels.
To generate these annotations, the authors design a markerless motion capture pipeline that integrates surface-based mesh tracking with subject-specific textured avatars, enabling precise reconstruction of individual body shapes and motions. They further introduce BigMac500, a curated benchmark subset for evaluating pose-informed action recognition, demonstrating that incorporating 3D pose descriptors leads to significant gains in mean average precision (mAP) across diverse visual backbones.

**Strengths:**

### **Strengths**

* The paper introduces a large-scale dataset that integrates detailed 3D pose–shape representations with action labels for macaques, addressing an important gap in current non-human primate research.
* The proposed annotation and reconstruction pipeline is innovative, combining markerless motion capture, subject-specific mesh tracking, and photometric texturing in a scalable way.
* The BigMac500 benchmark is a valuable contribution, demonstrating quantifiable improvements in action recognition performance when incorporating 3D pose descriptors.
* The annotation process **appears** well designed and yields annotations of good quality.
* The paper is simple and well written.
* The research problem is both interesting and challenging.
* The scale and diversity of the dataset is appealing.

**Weaknesses:**

### **Main Weaknesses**

The paper currently lacks sufficient detail about the dataset annotation pipeline. A clear, visual overview of the entire annotation process would be highly valuable—ideally presented as a dedicated figure (either in the main paper or, at minimum, in the appendix).

Secondly, the loss functions introduced for optimizing the annotations are not supported by ablation studies. Without these, it remains unclear which losses meaningfully contribute to the final annotation quality, and to what extent.

Moreover, there is **no quantitative evaluation of the annotation quality** beyond Table 2, which only reports IoU scores. This metric assesses 2D mask alignment but provides limited insight into the accuracy of the 3D surface models. It would be important to include additional metrics that reflect 3D fidelity and realism—such as *PSNR*, or *KL divergence*—as well as a measure of temporal consistency across frames. Since multiple losses are designed to optimize different parameters, it would be critical to demonstrate, both quantitatively and qualitatively, how these losses affect the reliability of the learned representations.

Finally, while the reported IoU of 0.844 is promising, it requires further clarification. Where do the remaining percentage points of error originate? A visualization comparing predicted and pseudo-ground-truth masks would help interpret this discrepancy—distinguishing between errors due to poor mesh fitting (undesirable), inaccurate pose estimation (undesirable), imperfect 2D masks (potentially acceptable), or minor border inaccuracies (acceptable).

### **Other Weaknesses**

* The paper provides no information about computational requirements. Details on runtime, GPU hardware, and annotation throughput should be included.
* The dataset was collected in controlled laboratory conditions, which may limit generalization to in-the-wild scenarios.
* The pose baseline used for prediction is not described. The paper should clarify the model architecture, input format, and training procedure.
* Loss ablations should be provided wherever feasible .

**Questions:**

### **Questions for the authors**

* I am not fully clear on what using “Real” rather than “Synthetic” 3D data concretely enables. Could you clarify the specific advantages it brings for model quality, bias reduction, or downstream generalization?
* How were the action labels produced in practice? Please describe who annotated them, the protocol, the ethogram mapping procedure, inter-annotator agreement, and any quality control. I could not find these details in the paper or the cited appendix.
* Can you provide evidence that the benchmark gains transfer to in-the-wild data? For example, a cross-domain test, few-shot adaptation, or evaluation on an external macaque or primate dataset.
* How well does the model personalize to individual monkeys? Concretely, how morphable is the 3D template, and what quantitative measures do you have on shape fitting accuracy across individuals?
* Do you quantify annotation uncertainty for the intermediate labels (segmentation, keypoints, identities) and analyze how these errors propagate to the mesh fitting and final pose vectors?
* How robust is the method to partial occlusion or missing views? Please report performance as a function of visible views, occlusion level, and view dropout, and describe any mechanisms that was used for handling this.

### **Summary comment**

The paper is strong and promising, but for a dataset and pipeline of this complexity, the current qualitative evidence is not sufficient. A set of targeted quantitative analyses would make the contribution much more convincing: ablations for each loss term, uncertainty propagation from 2D annotations to 3D fits, view-drop robustness, transfer to in-the-wild scenarios, and individual-specific morphability metrics. If provided, these results could substantially strengthen the work and its impact.

---

> ### Author Response · Authors · 2025-11-20
>
> We appreciate your comprehensive review and constructive comments. In accordance with the rebuttal character limits, we cite your comments in abbreviated form below and describe the changes we have already incorporated.
>
> > The paper ... detail about the dataset annotation pipeline. A clear, visual overview...
>
> Thank you for this suggestion! To clarify the distinction between the datasets and the pipeline, we added a new figure in the main paper.
>
> > Secondly, the loss ... extent. Moreover ... of the learned representations.
>
> We now include a loss ablation for the dynamic pose estimation in Appendix B5 to address the influence of individual loss terms, both quantitatively and qualitatively. As requested, we now include two more established measures to evaluate the surface estimation quality: MPJPE (absolute joint accuracy), and MPJTD (temporal smoothness of joint trajectories). Together, these measures address 3D skeletal alignment and the temporal smoothness of our surface reconstructions. We also note that the IoU in the multi-view scenario takes into account multiple viewpoint projections of the same surface that are informative about 3D surface fitting. We report all three measures for the qualitatively shown actions, and report substantial 3D alignment improvements and smoother trajectories against MAMMAL. We further extended the results for static frames for all single actions by reporting the MPJPE for the multi-view methods. In addition, we report both MPJPE and MPJTD for these actions over time with BigMac3D in the main text.
>
> > The paper provides ... should be included.
>
> Please note that we have now included the requested information in Appendix D.
>
> > The dataset was collected in ... scenarios.
>
> We agree that image-based detection and pose estimation models may face generalization challenges, as is common in animal-tracking tasks. Nevertheless, the 3D pose space generated by BigMac3D provides priors that can support reconstruction from in-the-wild images, similar to results in canine models.
>
> > The pose baseline ... procedure.
>
> We kindly refer to Appendix C, where we now clarify that the overall architectural structure remains the same for pose-only and visual-only recognition models.
>
> > ... “Real” rather than “Synthetic” ...
>
> Compared to synthetically generated 3D data, real multi-view 3D motion captures are not subject to biases introduced by the synthetic generation process that may fail to cover the full diversity of natural poses or even introduce unnatural ones. As a consequence, downstream models trained on synthetic 3D data risk inheriting these biases and may require additional filtering or domain adaptation. Moreover, when 3D supervision is derived from single-view inputs, the remaining depth ambiguities are resolved entirely through priors learned during synthetic generation, further amplifying them. This issue is expected to be pronounced for species with complex and flexible body postures, such as monkeys. In contrast, BigMac3D provides real multi-view recordings, which reduces these sources of bias and supports better generalization in downstream pose and behavior modeling. The inclusion of synthetic data arises out of the predicament that for animals ground-truth 3D data is limited.
>
> > How were the action labels produced ... appendix.
>
> We now describe the action label process in Appendix A3, and further quality controls in a dedicated new section, Appendix A5 that comprises the label-generation process for poses, bounding boxes, and masks. The agreement with other behavioral researchers regarding action recognition especially for the use case of single-view reconstruction in-the-wild is a valuable proposal for future work.
>
> > How well does the model personalize ... across individuals.
>
> To get a better intuition of how deformable the template is, the new Figure 8 in the Appendix presents an overview of the individual meshes for all subjects. We further quantify the ranges of the identity-related learnable parameters across subjects in Table 8.
>
> > Do you quantify annotation uncertainty ...
> > How robust ...
>
> We quantify the uncertainty of intermediate labels and incorporate this uncertainty into the mesh fitting process itself. In particular, the viewpoint-dependent losses are weighted by the detection confidence of an individual, and specific keypoints additionally by their confidence scores as produced by the keypoint estimation model. This is to reduce the effect of wrong keypoint estimates that produce stronger misalignments, as shown in the ablation Figure 9.
>
> We will address your remaining comments on realism, error propagation, and viewpoint dependency in our next response. We thank you again for your detailed and insightful review, and your confidence about our contribution overall. We hope to have addressed already a number of your suggestions, and we believe these changes have improved the manuscript.

---

> > ### Comment · Reviewer_Maxm · 2025-11-22
> >
> > Dear authors,
> >
> > Thank your for your response.
> >
> > Without clearly highlighting all the changes in the revised manuscript, it remains difficult to evaluate the full extent of the modifications. This is particularly challenging when authors only mention *“we added a new figure in the main paper”*. That said, the newly introduced Figure 3 is indeed a helpful addition.
> >
> > Thank you for adding the two additional metrics MPJPE and MPJTD. However, these metrics are not defined in the manuscript, except through references to prior work. Please define them mathematically in the context of your setup.
> >
> > Building on this, I appreciate the addition of Appendix B.5 on ablations. In your text, you write that *“This is apparent since without temporal loss the IoU scores remain relatively high even though the extremities can end up in weird rotations.”* This raises the question of why the paper does not include any pose-naturalness or joint-plausibility metrics or losses. Straightforward options exist. For example, joint-limit violation can be used when joint limits are known, or a pose-prior likelihood can be computed when a dataset exists. One example is [VPoser](https://github.com/nghorbani/human_body_prior) for humans. These metrics would provide a more principled evaluation rather than relying on a few manually selected examples where extremities end up in unnatural rotations.
> >
> > There is also an inconsistency in the IoU values reported. In Table X of the main paper you report an IoU of 0.844, while Table 10 in the Appendix (full model) reports an IoU of 0.855. Please clarify the difference between these two setups and how IoU is computed in each.
> >
> > In Appendix A.5, you mention “two other annotators” and a “video acquisition specialist”. Could you provide information about the time required for annotation, the total number of annotated keypoints, the annotation time per annotator, the distribution of reprojection errors in pixels, and the estimated annotation noise for 3D keypoints in millimeters? Even after the revision, the manuscript still does not present quantitative measures of annotation quality. Since the dataset is a central contribution, detailed annotation statistics are essential. I do not see how the dataset can be properly assessed without this information, especially prior to the fitting step.
> >
> > In Section A.5, you write:
> > > During mesh optimization, we additionally weight each camera viewpoint by its detection confidence. Furthermore, we also weight the contribution of individual keypoints using the confidence scores provided by HRNet-W48.
> >
> > Were these weighting mechanisms ablated? Detection confidence scores are typically not calibrated, which means they might not be reliable indicators for weighting. An ablation would be necessary here. I am also not aware of whether HRNet-W48 provides calibrated confidence scores, so an ablation of this component would also be informative.
> >
> > In your answer, you mention “The agreement with other behavioral researchers regarding action recognition especially for the use case of single-view reconstruction in-the-wild is a valuable proposal for future work.” While I agree with this point, it should explicitly appear in a limitations section of the main paper.
> >
> > Thank you for the new Figure 8 and Table 8. These additions are indeed interesting.
> >
> > Thank you again for your response. I look forward to the remaining clarifications regarding realism, error propagation, and viewpoint dependency. Even though the revision adds many missing details, the description of the annotation process remains too limited, and each additional detail raises further questions about the authors’ design choices. This reinforces my original rating.

---

> ### Author Response · Authors · 2025-11-30
>
> Dear reviewer,
>
> Thank you for your detailed response, we appreciate the time you invested in evaluating our revision. As in our previous reply, we quote your comments in abbreviated form, then address each point, clarify remaining issues, and provide the requested additional information.
>
> > Without clearly highlighting ... helpful addition.
> > Thank you for adding the two additional metrics ... define them mathematically in the context of your setup.
>
> We apologize for the inconvenience. We have now clearly highlighted all changes in the revised manuscript and provide mathematical definitions of the two additional metrics in Appendix B.5.
>
> > Building on this, I appreciate ... unnatural rotations.
>
> We agree that _“weird rotations”_ was not specific enough, and the manuscript now uses the term _“unnatural sequence of joint rotations.”_ We further clarify that we include a standard pose prior $L_P$ that constrains joint rotations, consistent with previous work on multi-view surface tracking in animals (e.g., MAMMAL; An et al., 2023). While we agree that this prior could be complemented or replaced with joint-limit constraints, such constraints would ideally be defined outside the axis–angle representation to more accurately reflect anatomical degrees of freedom. Thank you for raising this point; we will consider implementing such constraints for resolving surface tracking in more challenging single-view and static scenarios where they may be particularly beneficial.
>
> Please note that the inclusion of additional loss terms would still require appropriate weighting relative to the other regularization terms, informed by tracked examples.
>
> Regarding pose-naturalness and **VPoser**, we agree that such a model would be valuable. Since no equivalent dataset exists for macaques or other non-human primates, BigMac3D is specifically designed to address this gap. Consequently, a variational pose autoencoder (“MacPoser”) or pose prior trained on our fitted poses is indeed a natural next step, as mentioned in the future work section. Using a human SMPL pose prior would require nontrivial retargeting due to differences in skeleton structure, joint count, and template pose, introducing additional inaccuracies.
>
> >There is also an inconsistency in the IoU values reported. In Table X of the main paper you report an IoU of 0.844, while Table 10 in the Appendix (full model) reports an IoU of 0.855. Please clarify the difference between these two setups and how IoU is computed in each.
>
> Thank you for pointing this out. The values come from **different evaluation protocols**:
> - **Table 3 (main paper)** reports the **mean IoU across the first frame** of all single-subject actions.
>   This table evaluates only static frames because Animer+ lacks temporal regularization.
> - **Table 12 (Appendix; formerly Table 10)** reports evaluation on the **full temporal sequence** used in the ablation study.
>
> The caption of Table 12 now explicitly states:
> _“mean error metrics for the entire time sequence of the action shown in Figure 11. IoU scores are averaged over time and across cameras involved in the surface reconstruction.”_
> The IoU reported for the full model in the main paper (Table 2, second row, _Food Picking_) is consistent with this.
>
> > In Appendix A.5... Could you provide information about the time required for annotation, the total number of annotated keypoints, the annotation time per annotator, the distribution of reprojection errors in pixels, and the estimated annotation noise for 3D keypoints in millimeters? Even after the revision, the manuscript still does not present quantitative measures of annotation quality. Since the dataset is a central contribution, detailed annotation statistics are essential. I do not see how the dataset can be properly assessed without this information, especially prior to the fitting step.
>
> We apologize for the misunderstanding. In response to your initial request:
>
> >"Moreover, there is **no quantitative evaluation of the annotation quality** beyond Table 2, which only reports IoU scores. This metric assesses 2D mask alignment but provides limited insight into the accuracy of the 3D surface models. It would be important to include additional metrics that reflect 3D fidelity and realism—such as _PSNR_, or _KL divergence_—as well as a measure of temporal consistency across frames.">
>
> we focused on providing quantitative measures **beyond IoU**, adding two additional surface-tracking metrics in Table 3 (3D fidelity and temporal consistency).
>
> Following your updated request, we now also include in Appendix A.5:
>
> - number of annotated keypoints,
> - annotation time per annotator,
> - distribution of reprojection errors across keypoints,
> - estimated 3D annotation noise (in millimeters),
>
> among more detailed descriptions of the 3D labeling tool.
> We thank you for highlighting the importance of these details for dataset assessment.

---

> ### Author Response · Authors · 2025-11-30
>
> > In Section ... weighting mechanisms ablated? ... informative.
>
> We agree and have extended the ablation in Appendix B.6 to include **confidence weights**, which, although are generally not calibrated, still improve surface tracking quality.
>
> > In your ... appear in a limitations section of the main paper.
>
> We have now expanded the limitations section accordingly.
>
>  > Thank you again for your response. I look forward to the remaining clarifications regarding realism, error propagation, and viewpoint dependency.
>
> Regarding **error propagation and differences in IoU**, we note that SAM 2 pseudo–ground truth may occasionally include background regions or exclude parts of the animal. We extend the qualitative examples in Appendix B.7 by presenting such masks and a case where inconsistent multi-view detections affect tracking performance, even if not severely. We also emphasize this in the limitations and future work section:
> “More challenging multi-individual scenes would benefit from view-consistent detection agreement, and SAM 2 silhouettes could be further improved with a dedicated mask-quality discriminator.”
>
> To address **viewpoint dependency**, we extended Appendix B.6 with an ablation on the number of cameras used. As expected, surface tracking accuracy improves across all metrics with additional viewpoints, especially when increasing from two to three cameras.
>
> We would like to conclude with discussing the **realism** of reconstructed actions. We want to thank you for bringing this to our attention. While we provide textured avatars, the texture represents a mean appearance across scenes for a specific individual and does not adjust to per-frame lighting conditions. Since our photometric objective directly penalizes pixel-wise differences, PSNR or KL divergence would capture similar information. Instead, we considered perceptual similarity metrics (e.g., [LPIPS](https://github.com/richzhang/PerceptualSimilarity)) between the original and the rendered image, but applying them meaningfully would require expanding the color parameters over time and, ideally, perceptual models aligned with monkey neural or behavioral responses to assess realism of actions accurately.
>
> Thank you again for your thoughtful comments. They have led us to strengthen the manuscript, incorporate additional quantitative analyses, and identify valuable directions for further research.

---

### Official Review · Reviewer_nfdd · 2025-11-01

**Soundness:** 2
**Presentation:** 3
**Contribution:** 2
**Rating:** 6
**Confidence:** 3

**Summary:**

This paper presents BigMac3D, a large multi-view video dataset of rhesus macaques with dense surface-based 3D pose/shape reconstructions and action labels (an ethogram). The recordings come from 16 calibrated cameras in a neuroscience laboratory and cover >750 action scenes and a derived action-recognition benchmark (BigMac500) of $~\sim$8k labeled videos. The authors build subject-specific textured avatars by adapting a high-quality macaque template mesh and introduce processing/optimization improvements (symmetric time loss, cropped differential rendering, integrated texture) to make large-scale surface reconstruction tractable.

**Strengths:**

Impactful dataset: fills an important gap for non-human primate research by linking dense 3D surface reconstructions with behavior labels.

Practical pipeline: believable and reproducible-in-principle mesh-fitting and rendering pipeline that scales to hundreds of scenes.

Demonstrated utility: empirical evidence that 3D pose/shape descriptors can improve action recognition across multiple visual backbones.

Rich annotation set: identities, segmentation masks, 2D keypoints, calibrated views, per-frame poses and action labels.

**Weaknesses:**

Ethical / animal welfare documentation (major). The recordings come from a neuroscientific laboratory and involve captive macaques. I cannot find any explicit statement of oversight, animal-use committee (e.g., IACUC or institutional equivalent) approval, or details of welfare protocols. The paper says recordings were “not induced by humans except feeding,” but ethical approval and animal care protocols must be stated explicitly.

Statistical significance and variance. Report standard deviations / confidence intervals and number of seeds for all recognition results (Tables 3-4) and any other ML experiments. If training is deterministic (unlikely), explain why; otherwise run multiple seeds and report variability. Also provide basic compute footprints (GPU types, training time per model, FLOPs if possible) so the community can compare methods fairly.

Annotation reliability. How were action labels assigned? How many annotators per clip? Multi-label or single-label? What was the inter-annotator agreement (kappa / % agreement)? Actions like “aggression,” “dominance display,” or “anxiety” can be subjective—please document.

Dataset bias & generalization. The dataset contains eight male macaques in one lab. Discuss how sex, age, enclosure, and captive vs. wild behavior may affect generality. Are there plans to extend diversity (females, juveniles, different environments)?

Privacy / safety considerations. While this is animal data, the authors should discuss potential misuse (e.g., tracking in the wild or exploiting behavior data) and state any restrictions on dataset use if applicable.

**Questions:**

Refer to questions

**Details Of Ethics Concerns:**

Animal ethics review.

---

> ### Author Response · Authors · 2025-11-20
>
> Thank you for your constructive review and support for our work!
> In the following, we would like to address your raised concerns:
>
> >Ethical / animal welfare documentation (major). ...
>
> Please let us refer you to the Appendix, where we provide more information on the ethics approval and welfare of the animals. Since this is a critical issue, we have now also incorporated an ethics statement at the end of the main paper.
>
> > Privacy / safety considerations. While this is animal data ...
>
> This statement further addresses regulatory compliance, privacy concerns, and discusses potential misuse of the data beyond animal-welfare contexts.
>
> > Statistical significance and variance. Report standard deviations / confidence intervals and number of seeds for all recognition results (Tables 3-4) and any other ML experiments. If training is deterministic (unlikely), explain why; otherwise run multiple seeds and report variability.
>
> The action recognition benchmark results now include multiple training seeds, along with standard deviations / confidence intervals as requested. As initially observed, the action-recognition performance in Tables 3–4 (now Tables 4–5) continues to show substantial improvements when using pose features derived from our surface reconstructions.
>
> > Also provide basic compute footprints (GPU types, training time per model, FLOPs if possible) so the community can compare methods fairly.
>
> We now report GPU type (NVIDIA GeForce RTX 3090), annotation throughput for surface tracking, as well as the range of FLOPs and training times for action recognition models in the Appendix D.
>
> > Annotation reliability. How were action labels assigned? How many annotators per clip? Multi-label or single-label? What was the inter-annotator agreement (kappa / % agreement)? Actions like “aggression,” “dominance display,” or “anxiety” can be subjective—please document.
>
> The action labels were assigned based on the behavioral descriptions in our ethogram, which further specifies categories using distinct behavioral expressions reported in previous ethological studies of macaques. Detailed category documentation can be found in Appendix A3. Annotations are inherently multi-label, since monkeys may exhibit multiple behaviors simultaneously; this is now stated more explicitly in the “Actions and Labels” section. Annotations were produced by one annotator in consultation with an expert who works daily with the monkeys for an initial set of labels. While formal inter-annotator agreement metrics were not collected, future work could incorporate cross-validation with additional behavioral researchers, especially for tracking scenarios extending our laboratory conditions. We now explicitly acknowledge this limitation in the paper.
>
> > Dataset bias & generalization. The dataset contains eight male macaques in one lab. Discuss how sex, age, enclosure, and captive vs. wild behavior may affect generality. Are there plans to extend diversity (females, juveniles, different environments)?
>
> Future work could also include more diverse recordings of monkey subjects, since the reconstructed animals were 5–7 years old at the time of recording. It is true that the generality of the 2D pose and detection models is, to some extent, limited by the environmental conditions and would require domain adaptation for use in different settings, which is a common issue and purpose of interactive labeling and training toolboxes such as DeepLabCut. However, an integral use case of BigMac3D's pose space is to generalize to and regularize the retrieval of ambiguous 2D macaque poses for in-the-wild recordings or single-view surface estimation models, as demonstrated for dogs with BARC and RGBD-Dog in prior work (Rüegg et al., 2022 and Kearney et al., 2020). Since this is a major avenue we are actively pursuing, we now emphasize this point more clearly in the future-work discussion.
>
> We believe that the changes have strengthened the manuscript, and we hope that these revisions adequately address your concerns and strengthen your confidence in our contribution.

---

### Author Response · Authors · 2025-12-01

We sincerely thank all reviewers, nfdd, Maxm, Q757, and 6cL8 for the time and care they dedicated to evaluating our work, and we regret that the discussion ended prematurely. Their constructive comments, suggestions, and detailed questions have all directly contributed to strengthening and clarifying our manuscript. We greatly appreciate the reviewers’ suggestions for making our dataset more comprehensive. As the first dataset to provide realistic dynamic surface reconstructions of non-human primates, we found their perspectives invaluable for improving clarity and documentation. We view this process as a collaborative effort and are grateful for their engagement. We also acknowledge and welcome the suggestions regarding potential extensions to the surface-tracking approach, and we are encouraged that the reviewers did not question but rather affirmed our core findings. We were pleased to see several consistent points of positive feedback across the reviews:

- BigMac3D fills a major gap in non-human primate and behavioral research coupling realistic 3D surface reconstructions with action recognition.
- The richness and completeness of the dataset were repeatedly emphasized, being described as _"unprecedented"_.
- Reviewers highlighted the practical, believable, and scalable surface-tracking pipeline.
- There was clear recognition of the empirical gains in action recognition enabled by surface-based 3D pose descriptors.

The reviewers’ constructive input has helped us substantially improve the manuscript. In response, we have now included:

- A clear overview of the reconstruction pipeline
- Two additional measures that reaffirm reconstruction quality and 3D fidelity
- Computational requirements
- More detailed descriptions of the annotations, annotation process, quality control steps and uncertainty estimates of keypoints in 2D and 3D
- An overview and statistics of individual shape adaptations
- Ablations on loss weighting factors, confidence weights, and the number of viewpoints
- Extended qualitative analysis of failure cases, including the presentation of labels for alignment in the Supplementary Video
- A revised future work and limitations section
- A dedicated ethics statement in the main text
- Clarifications regarding data splits and a short introduction to surface models
- Cofindence intervals from multiple training runs for the action recognition benchmark, and distributions of action sequence lengths in the dataset
- A figure showing the spatial camera ID distribution of the recordings

We have highlighted all changes in the manuscript, and we believe they have substantially strengthened its clarity, rigor, and comprehensiveness. We are grateful that all reviewers recognized the contribution and potential impact of our work.

Once again, we thank the reviewers for their encouragement and for helping us improve the manuscript.

---

### Meta-Review · Area_Chair_SN72 · 2026-01-05

**Summary:**

This paper introduces BigMac3D, a dataset coupling dense surface-based 3D reconstructions of rhesus macaques with ethogram-aligned action labels, alongside the BigMac500 benchmark.

Reviewers acknowledged the dataset's scale, richness, and relevance, noting that it fills a gap for non-human primate research and demonstrates empirical gains in action recognition from surface-based 3D pose descriptors. The surface-tracking pipeline was viewed as practical and scalable.

Initial concerns focused on insufficient documentation of ethics, annotation quality, reconstruction fidelity, reproducibility, and clarity of the pipeline and dataset splits. Through rebuttal and discussion, the authors substantially addressed many of these issues: they added a clear pipeline overview, expanded ethics statements, detailed annotations and quality control, provided computational requirements, introduced additional quantitative 3D metrics (MPJPE, MPJTD), loss and confidence weight ablations, camera and sequence statistics, and clarified BigMac3D vs. BigMac500 and data splits.

Several concerns remain only partially resolved, particularly around the depth of annotation reliability analysis, pose naturalness metrics, and broader generalization beyond the controlled lab setting. Reviewer opinions ranged from enthusiastic acceptance to reservations about dataset validation rigor, but several critiques were mitigated during discussion.

**Reviewer Concerns:**

See above.

**Reviewer Scores:**

Some reviewers may raise their scores after the rebuttal clarifies and addresses their concerns, while others, who already gave high initial scores, may not.

---

### Decision · Program_Chairs · 2026-01-26

Accept (Poster)